# Joint Prompt Optimization of Stacked LLMs using Variational Inference

**Alessandro Sordoni**[ab*]  **Xingdi Yuan**[a]  **Marc-Alexandre Côté**[a]  **Matheus Pereira**[a]

**Adam Trischler**[a]  **Ziang Xiao**[a]  **Arian Hosseini**[b]  **Friederike Niedtner**[a]  **Nicolas Le Roux**[ab]

Microsoft Research Montréal[a]        MILA[b]

## Abstract

Large language models (LLMs) can be seen as atomic units of computation mapping sequences to a distribution over sequences. Thus, they can be seen as stochastic language layers in a language network, where the learnable parameters are the natural language prompts at each layer. By stacking two such layers and feeding the output of one layer to the next, we obtain a Deep Language Network (DLN). We first show how to effectively perform prompt optimization for a 1-Layer language network (DLN-1). Then, we present an extension that applies to 2-layer DLNs (DLN-2), where two prompts must be learned. The key idea is to consider the output of the first layer as a latent variable, which requires inference, and prompts to be learned as the parameters of the generative distribution. We first test the effectiveness of DLN-1 in multiple reasoning and natural language understanding tasks. Then, we show that DLN-2 can reach higher performance than a single layer, showing promise that we might reach comparable performance to GPT-4, even when each LLM in the network is smaller and less powerful. The DLN code is open source.[1]

## 1 Introduction

The size of large language models (LLMs) has grown significantly over the last few years, mainly because of emerging capabilities [5, 31], but at considerable technical and societal costs [49, 2, 4]. Recent efforts have focused either on learning smaller models matching the abilities of larger ones on some tasks using distillation [43, 36, 29, 13], or offloading part of the computation to other dedicated components [28, 22, 25, 18]. In the latter case, this is done through carefully crafted instructions to retrieve the necessary information from these additional modules [48, 41, 6, 54, 24].

Instruction-tuned LLMs map an input sequence to a distribution over output sequences conditioned on an instruction, or *prompt*. In this paper, we view such LLMs as stochastic language layers, whose learnable parameters are the prompts. Multiple layers can be stacked to form a Deep Language Network (DLN) whose learnable parameters are the prompts associated to each layer. Specifically, each layer uses a template to organize both its prompt and the inputs coming from the layer below into a single sequence before producing the output (see Figure 1). This layering induces a learnable decomposition of the task into a series of smaller sub-tasks, each of which might be more easily solvable by an LLM. This view shares similarities to recent works that chain LLM calls [41, 6, 54]. In this work, we move towards integrating learnable components in the pipeline: each prompt can be learned to maximize the final objective.

---

[*]Corresponding author: alsordon@microsoft.com
[1]Code: https://github.com/microsoft/deep-language-networks

37th Conference on Neural Information Processing Systems (NeurIPS 2023).

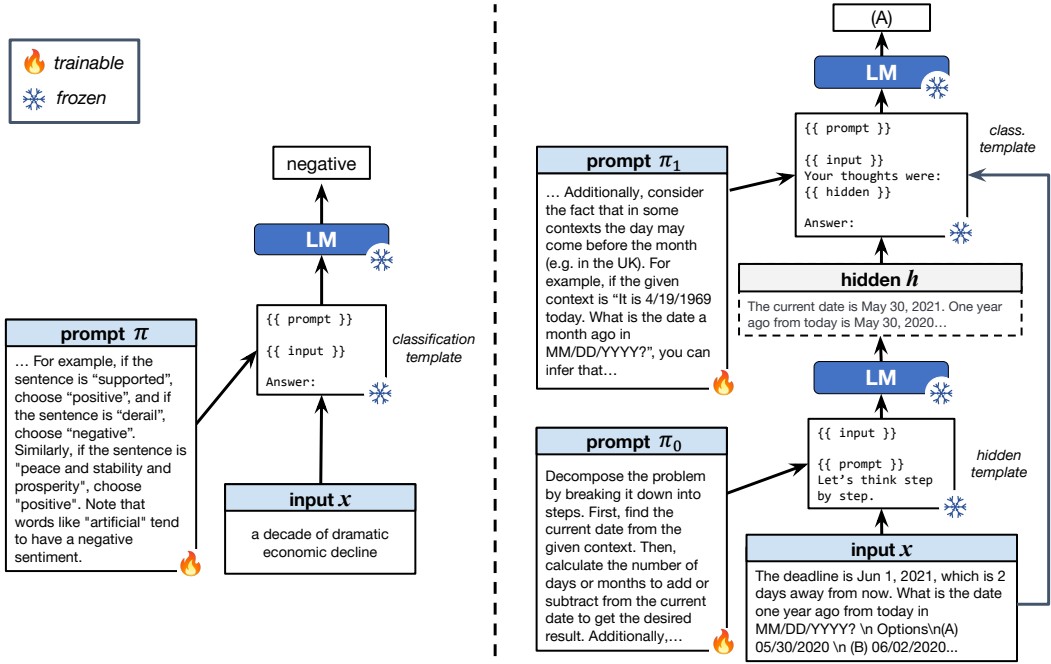

Figure 1: *Left*: An illustration of a DLN-1 performing a sentiment analysis task: `input` and the trainable `prompt` are merged using a template and fed to the LM for answer generation. *Right*: a DLN-2 with a residual connection, performing the date understanding task: two prompts need to be learned. In this example, the hidden template extends Chain-Of-Thought [48] with a learnable prefix; we consider the output of the first layer, `hidden`, as a *latent variable h*. We use *variational inference* to learn $\pi_0, \pi_1$. Templates can be considered as an hyperparameter of the network.

We first show how to perform prompt optimization in a shallow 1-layer language network (DLN-1) which parametrizes a distribution $p_{\text{LM}}(y|x, \pi)$, where $x$ and $y$ are string input and output respectively, and $\pi$ is the learnable prompt (Figure 1, *left*). Our prompt optimization techniques extend the Automatic Prompt Engineer (APE) procedure from Zhou et al. [57]. We show how our prompts can include a verbalization of difficult examples from the task: the final prompts are a combination of instruction directives, akin to zero-shot learning [17], and task examples, akin to in-context learning [20]. This significantly improves downstream performance, surpassing APE on several tasks.

Then, we show how to train a 2-layer DLN (DLN-2), which parametrizes a probability distribution:

$$p_{\text{DLN-2}}(y|x) = \sum_h p_{\text{LM}}(y|h, x, \pi_1)\, p_{\text{LM}}(h|x, \pi_0)\,,$$

where $h$ is the string output of the first LLM layer (Figure 1, *right*). We consider $h$ as a latent variable: to maximize the marginal log-likelihood, we formulate a variational inference algorithm that uses an approximate posterior over $h$. Note that this formalism easily encompasses more than two layers.

Considering outputs of the hidden language layers as latent variables allows us to encompass various established prompting methods, such as Chain-Of-Thought (CoT) [48] and self-consistency (SC-CoT) [46]. Particularly, CoT can be seen as a particular DLN-2 with the first layer prompt set to ''Let's think step by step'' and the second layer prompt set to ''The answer is''; we can either learn such prompts or learn a supplement to those as in Figure 1. SC-CoT can be seen as marginalizing over CoT strings sampled from a task-agnostic prior: when using the template in Figure 1 (*right*), our method generalizes this perspective by learning a task-specific prior distribution over successful CoTs.

The rest of the paper is organized as follows. First, we provide an interpretation of LLMs as shallow networks, drawing a number of analogies with standard parametric and non-parametric models and explaining how best to train them. After exploring their limitations, we propose to stack two such

LLMs to form a DLN-2. We show how they can be trained using a form of variational inference, then demonstrate their performance on a series of reasoning and language understanding tasks.

## 2 One-Layer Language Networks

A pre-trained LLM with frozen weights might be thought of as a complete *function class* indexed by prompts. The output $y$ of an LLM for an input $x$ can be modulated through a prompt $\pi$ by feeding a combination of $\pi$ and $x$ to the LLM. Hence, from a low-level perspective, the function class of an LLM is defined by its architecture, i.e., its depth, number of heads, context size, etc., and training happens at the parameter level. It is data and compute intensive and should be done rarely. From a high-level perspective, the function class of an LLM is defined by the pre-trained model chosen (LLAMA [44], text-davinci-003, GPT-4, etc.), and training happens by fine-tuning the model or by choosing the prompt.

There are two ways of optimizing an LLM at the prompt level. The first one is *prompt engineering*, a parametric optimization, where the optimization space is independent of the size of the dataset. Because this optimization usually happens in discrete space, gradient-based techniques do not apply and most efforts rely on a combination of random or local search and human heuristics [21, 55]. The second one is *in-context learning* (ICL), a non-parametric optimization technique where the solution is a direct function of a subset of examples [5, 20]. This approach works well for few-shot learning but scaling it to larger datasets has both performance and computational issues. We shall now generalize previous work in discrete prompt optimization [57, 21] with the ultimate goal of learning a set of prompts in a language network.

### 2.1 Language Layers

We use *language layer* to refer to a (stochastic) computation that takes as input a string $x$ and outputs a string $y$. This computation is modulated by another string, $\pi$, generally called a *prompt* or *instruction* [57]. The string transduction is performed by an operator LM, by feeding $x$ and $\pi$ as context and generating a continuation $y$. *Templates* describe the way $x$ and $\pi$ are combined prior to being fed to the LM operator. These are functions that accept strings as variables and output a string. We will denote such templates with this font T. A simple forward template F is the concatenation, i.e. $\texttt{F}(x, \pi)$ = "$\{\pi\}\{x\}$". We also explore more complex ones, examples of which can be seen in Figure 1.

Given an input $x$, a prompt $\pi$, and a template F, a language layer defines a probability distribution $p_{\texttt{LM}}(y|\texttt{F}(x, \pi))$ over output strings $y$ as computed by the LM. In the next section, we describe a generic framework for optimizing the weights $\pi$ for a language layer.

### 2.2 Prompt Optimization: Improved APE

Because the search for the best prompt happens over a discrete space, we will rely on a local search method, using an LLM to implement a distance measure between prompts. The procedure can be seen as an extension of Automatic Prompt Engineer (APE), recently proposed by Zhou et al. [57], and will serve as a stepping stone towards introducing our algorithm for training deep language networks. Our prompt optimization algorithm can be structured as follows:

1. Given the current prompt $\pi$ and a current batch of examples $\{x, y\}$, generate $N$ "local" candidates $\pi^1, \ldots, \pi^N$ using a prompt *proposal* distribution;

2. Score each candidate using a (potentially stochastic) *scoring* function $s$, then choose $\pi = \arg\max_{\pi^n} s(\pi^n)$.

**Prompt Proposal** Local search algorithms assume a distance measure between inputs to crawl the search space. In this setting, we rely on LLMs to generate local modifications to the prompts. Our prompt proposal distribution takes as conditioning information i) the batch given as input to the layer, ii) its corresponding output $\{x, y, \hat{y}\}$, and iii) the current prompt $\pi$. The proposal distribution $p_{\texttt{LM}}(\pi^n|\texttt{B}_\pi(\{x, y, \hat{y}\}, \pi))$ wraps this information using a particular "backward" template $\texttt{B}_\pi$, which can be found in Appendix D. This approach is similar to the instruction template used by Zhang et al. [55], with the exception that we also integrate information about the model's own predictions, which

---
**Algorithm 1** One-Layer Language Network (DLN-1) Training Algorithm
---
**Require:** $\hat{y} \sim p_{\mathtt{LM}}^t(y|c)$                  ▷ generates a completion of prefix $c$ with temperature $t$
**Require:** $\log p_{\mathtt{LM}}(h|c)$                             ▷ return log-prob of $h$ following $c$
**Require:** $N$: prompt samples, $I$: iterations, $\mathcal{D}$: dataset
**Require:** F: template for the inference/forward pass
**Require:** $\mathtt{B}_\pi$: template for prompt proposal/backward pass.
  1: Initialize $\pi$ with a task description or empty
  2: **for** $i$ in $[1, I]$ **do**
  3:     $x, y \sim \mathcal{D}$                            ▷ Sample minibatch
  4:     $\hat{y} \leftarrow p_{\mathtt{LM}}^0(y|\mathtt{F}(x, \pi))$                ▷ Do inference pass
  5:     $\pi^1, \ldots, \pi^N \sim p_{\mathtt{LM}}^{0.7}(\pi|\mathtt{B}_\pi(\{x, y, \hat{y}\}, \pi))$     ▷ Sample $N$ candidate prompts
  6:     $s^1, \ldots, s^N \leftarrow \log p_{\mathtt{LM}}(y|\mathtt{F}(x, \pi^n))$         ▷ Score all prompts
  7:     $\pi \leftarrow \arg\max_{\pi^n}\{s^1, \ldots, s^N\}$       ▷ Select prompt with best score
  8: **end for**
---

we found to empirically help performance given that the model tends to propose prompts that correct its own errors. We sample from the prompt proposal distribution to generate a set of $N$ prompts. A particularly important aspect is ensuring the diversity of the candidate pool $\pi^1, \ldots, \pi^N$. We devise several strategies to improve the diversity and the usefulness of the candidate samples in Section 4.

**Prompt Selection** Once a set of $N$ prompts has been generated, we use a scoring function to select the updated prompt. We assume access to the log-likelihoods of the LM operator and we rank the candidate prompts to maximize data log-likelihood $\pi = \arg\max_{\pi^n} \log p_{\mathtt{LM}}(y|\mathtt{F}(x; \pi^n))$. In practice, we normalize this log-probability by the length of the output string. While we focus on that metric in this work, there is no restriction on the scoring function that can be used. We use backtracking to increase the robustness of our selection mechanism, as well as a memory of well-performing prompts for efficiency. We present both strategies in Section 4. The sketch of a 1-layer prompt optimization algorithm is described in Algorithm 1, ignoring backtracking and memory for simplicity.

The results of our prompt optimization may be found in Table 1 and will be discussed in detail in Section 5.2. We now turn to extending prompt optimization to architectures with two layers.

## 3 Two-Layer Deep Language Networks (DLN-2)

The natural extension of DLN-1 is DLN-2, in which language layers are stacked, i.e. the output of the first language layer is the input to the second one. A 2-layer network induces a distribution over outputs of the form:

$$p_{\mathtt{DLN\text{-}2}}(y|x) = \sum_h p_{\mathtt{LM}}(y|\mathtt{F}_r(h, x, \pi_1))p_{\mathtt{LM}}(h|\mathtt{F}(x, \pi_0)) \tag{1}$$

where $h$ is a latent variable that potentially makes it easier to explain the target $y$. The output layer is also conditioned on $x$ through $\mathtt{F}_r$, forming a residual connection (Figure 1). This formulation is reminiscent of past work using latent language representations to guide document summarization [26]. In our case, however, the encoding/decoding distributions are parameterized by natural language prompts $\Pi = \{\pi_0, \pi_1\}$, and we do not assume access to the LLM parameters.

While this architecture has more expressive power than a shallow language network, the prompt optimization problem becomes harder now that we have to *jointly* search over both $\pi_0$ and $\pi_1$. Doing random search in this space is impractical [8] and manual tuning of the weights is also exponentially harder than with a single prompt. We turn to variational inference to address this issue.

### 3.1 Variational Inference Objective

The layerwise decomposition of our system allows us to leverage tools from approximate inference in probabilistic models to learn $\Pi$. In particular, we propose to use variational inference to learn $\Pi$ [3, 16]. We posit an approximate posterior $q(h)$ over the latent variable $h$, and bound the marginal

log-likelihood of $y$ given $x$ by computing the ELBO:

$$\log p_{\texttt{DLN-2}}(y|x) \geq \sum_h q(h) \left[\log p_{\texttt{LM}}(y|\mathsf{F}_r(h, x, \pi_1)) p_{\texttt{LM}}(h|\mathsf{F}(x, \pi_0))\right] + H\left[q(h)\right], \qquad (2)$$

which allows us to decompose the optimization over $\Pi$ in two independent optimization problems, over both $\pi_0$ and $\pi_1$:

$$\pi_0^* = \arg\max_{\pi_0} \sum_{x,\, h} w_h \log p(h|\mathsf{F}(x, \pi_0)), \quad \pi_1^* = \arg\max_{\pi_1} \sum_{(x,y),\, h} w_h \log p(y|\mathsf{F}(h, x, \pi_1)) . \qquad (3)$$

The search over $\pi_1$ is identical to the prompt optimization described in Section 2, with the difference that the inputs now depend on the approximate posterior samples $h$ in addition to the inputs $x$. The search over $\pi_0$ uses a similar strategy but uses the posterior samples $h$ as targets, instead of $y$.

Although this bound allows us to decompose the optimization w.r.t. $\Pi$, it is only useful if it is close to the true value. Since its looseness is the KL divergence between $q(h)$ and the true posterior, $\text{KL}(q(h)||p(h|y, x))$: we need to find an approximate posterior $q$ closely matching the true posterior. In what follows, we specify how we parametrize the approximate posterior and how we tighten the approximation via posterior sharpening.

**Hidden Proposal** We will also be using an LLM to sample candidate hidden states from $q(h)$. Unless specified, for simplicity, we use the same LM operator used in our language layers. The approximate posterior can condition on arbitrary amount of information but especially useful might be to condition on the true target $y$. If it conditions on the hidden state coming from the "forward pass", $\hat{h} \sim p_{\texttt{LM}}(h|\mathsf{F}(x, \pi_0))$, then $q_{\texttt{edit}}(h) = p_{\texttt{LM}}(h|\mathsf{B}_h(\hat{h}, y, \pi_1))$. $\mathsf{B}_h$ is a specifically tailored hidden proposal template (Appendix D). $q_{\texttt{edit}}$ performs a sort of edit operation, where the LM is tasked to rewrite the hidden variable $\hat{h}$ given some extra knowledge of the ground-truth label $y$ and of $\pi_1$. Alternatively, we can set the posterior to be equal to the prior, i.e. $q_{\texttt{pri}}(h) = p_{\texttt{LM}}(h|\mathsf{F}(x, \pi_0))$, or the prior with additional information about the label $y$, $q_{\texttt{pri+}}(h) = p_{\texttt{LM}}(h|\mathsf{B}_y(x, \pi_0, y))$ (Appendix D). This amounts to re-computing the hidden state knowing privileged information about the label. We found most effective to sample hidden states from a mixture of $q_{\texttt{pri}}$ and $q_{\texttt{pri+}}$.

**Posterior Sharpening** Given the absence of learnable parameters in $q(h)$, the induced approximate posterior might still be far from the true posterior. To bridge this gap, we reweigh each sample $h^i$ based on its probability under the true posterior distribution. More precisely, we compute $\tilde{w}^i = \log p_{\texttt{LM}}(y|\mathsf{F}_r(h^i, x, \pi_1)) + \log p_{\texttt{LM}}(h^i|\mathsf{F}(x, \pi_0))$, then assign to each $h_i$ the probability $w_i = \exp(\alpha \tilde{w}_i)/\sum_j \exp(\alpha \tilde{w}_j)$, where $\alpha$ is a tunable temperature parameter that controls the entropy of the posterior weights. The full algorithm for training a DLN-2 is presented in Algorithm 2.

## 4 Practical Instantiation

Although our method aim to learning prompts in stacked LLM architectures, we do rely on a good amount of prompt engineering for our templates. Hereafter, we detail some choices that were fundamental to make our approach work in practice.

**Proposal Diversity** To ensure a diversity of the samples for both the prompt proposal distribution, we found helpful to use two strategies. The first is to modify the backward templates $\mathsf{B}_\pi$ before drawing a sample from the proposal distribution $p_{\texttt{LM}}(\pi)$. To achieve so, we parametrize the basic templates with a ``{message}'' variable that we instantiate from a pool of hand-written instructions, that describe different behaviors the model should follow to propose new $\pi$, e.g. ``shorten the previous instruction'', ``give useful examples'', etc. These can be interpreted as *meta-instructions*, i.e. high-level directives that inform the model on how to create a better instruction for the task, and extend instruction-induction templates used in [12, 55]. These can be found in Appendix D. In the future, we could envision to extend learning to these instructions. In the case of $\mathsf{B}_\pi$, they could function as parameters for a *prior* over the weights of the DLN. The second strategy to ensure more diversity is that we instantiate $\mathsf{B}_\pi$ with a different random subset of examples in the current batch, before drawing each sample $\pi^n$. This effectively modifies the generation context for each sample $\pi^n$.

**Learning In-Context Learning** One strategy we found particularly effective is to integrate in the pool of meta-instructions an additional instruction that asks the LM to give useful examples to improve

**Algorithm 2** Two-Layer Deep Language Network (DLN-2) Training Algorithm

1: $\hat{y} \sim p_{\texttt{LM}}^t(c)$        ▷ generates a completion of prefix $c$ with temperature $t$
2: $\log p_{\texttt{LM}}(h|c)$        ▷ return log-prob of $h$ following $c$
3: $N$: prompt samples, $K$: posterior samples, $I$: iterations, $\mathcal{D}$: dataset
4: $\texttt{F}, \texttt{F}_r$: templates for the inference/forward pass without and with residual connection
5: $\texttt{B}_\pi, \texttt{B}_h$: templates for prompt and hidden proposal/backward pass.
6: Initialize $\pi_0, \pi_1$ with a generic sentence or task description
7: **for** $i$ in $[1, I]$ **do**
8:      $x, y \sim \mathcal{D}$        ▷ Sample minibatch
9:      $\hat{h} \leftarrow p_{\texttt{LM}}^0(\texttt{F}(x, \pi_0))$        ▷ Do inference pass, first layer
10:     $\hat{y} \leftarrow p_{\texttt{LM}}^0(\texttt{F}_r(\hat{h}, x, \pi_1))$        ▷ Do inference pass, second layer
11:     $h^1, \ldots, h^K \sim p_{\texttt{LM}}^{0.7}(\texttt{B}_h(\hat{h}, x, y, \pi_1, \pi_0))$        ▷ Sample $K$ posterior proposals for $h$
12:     $\alpha^1, \ldots, \alpha^K \leftarrow \log p_{\texttt{LM}}(h^k|\texttt{F}(x, \pi_0))$        ▷ Compute prior log-probs for all $h^k$ samples
13:     $\beta^1, \ldots, \beta^K \leftarrow \log p_{\texttt{LM}}(y|\texttt{F}_r(h^k, x, \pi_1))$        ▷ Compute log-likelihoods for all $h^k$ samples
14:     $q^k \leftarrow \exp(\alpha^k + \beta^k)/(\sum_k \exp(\alpha^k + \beta^k))$        ▷ Compute normalized posterior weights
15:     $h^* \leftarrow \arg\max_h\{q^1, \ldots, q^K\}$        ▷ Select best posterior sample
16:     $\pi_0^1, \ldots, \pi_0^N \sim p_{\texttt{LM}}^{0.7}(\texttt{B}_\pi(\{x, \hat{h}, h^*\}, \pi_0))$        ▷ Sample $N$ candidate prompts for $\pi_0$
17:     $\pi_1^1, \ldots, \pi_1^N \sim p_{\texttt{LM}}^{0.7}(\texttt{B}_\pi(\{\hat{h}, \hat{y}, y\}, \pi_1))$        ▷ Sample $N$ candidate prompts for $\pi_1$
18:     $s_0^1, \ldots, s_0^N \leftarrow \sum_k q^k \log p_{\texttt{LM}}(h^k|\texttt{F}(x, \pi_0^n))$        ▷ Compute ELBO for all prompts $\pi_0^n$
19:     $s_1^1, \ldots, s_1^N \leftarrow \sum_k q^k \log p_{\texttt{LM}}(y|\texttt{F}_r(h^k, x, \pi_1^n))$        ▷ Compute ELBO for all prompts $\pi_1^n$
20:     $\pi_0 \leftarrow \arg\max_{\pi_0^n}\{s_0^1, \ldots, s_0^N\}$        ▷ Select prompt $\pi_0$ with best score
21:     $\pi_1 \leftarrow \arg\max_{\pi_1^n}\{s_1^1, \ldots, s_1^N\}$        ▷ Select prompt $\pi_1$ with best score
22: **end for**

its current prompt $\pi$. Empirically, we observed that this allows the model to sample candidate prompts $\pi^n$ that contain synthetic examples for the task, embedded in natural language. Examples of this interesting behavior can be found in Appendix F. We found that this behavior is particularly interesting as the resulting prompts often perform better than standard ICL. We hypothesize this is due to both *i)* the "verbalization" of the example in the prompt, which modifies the dataset syntax into a more suitable one, and *ii)* the fact that the model can dynamically select which examples are most important to integrate in the prompts, given the errors made during training. Therefore, we suspect that DLN achieves a similar effect to recent techniques that select important examples for ICL [20, 35, 40, 18, 47] with the improvement of naturally conditioning the selection on the end task performance via end-to-end training.

**Backtracking and Memory** Optimization of both DLN-1 and DLN-2 is challenging due to the fact that we do not have gradient information and we sample a restricted set of candidates $\pi^n$ at each optimization step due to computational reasons. We deploy multiple strategies to allow the network to be robust to sampling/selection errors. First, we include the current prompt $\pi$ into the set of candidate prompts to be scored at the current iteration $\pi^n$. This allows the model to not take the step if the previous prompt performed better. Second, we keep a memory of $M = 5$ best prompts found by tracking validation set performance.

**Exploration Reward** When training a DLN-2, we empirically observed that the first layer prompt $\pi_0$ was updating very slowly. Due to the fact that the approximate posterior shares templates with the prior used in the forward pass, the posterior samples $h^i$ are close to $\hat{h}$ and the maximizer of Equation (3) remains $\pi_0$. To address this issue, we add to the scores of each candidate prompt an exploration reward that is proportional to the negative log-probability of those $\hat{h}$ that led to an incorrect prediction: $r = -\lambda \log p_{\texttt{LM}}(\hat{h}|\texttt{F}(x, \pi^n))$, if $\hat{y} \neq y$. This encourages the model to both find prompts that maximize the log-probability of high-probability posterior samples and at the same time minimize the log-probability of prior samples that led to incorrect predictions. We anneal $\lambda$ to 0 during training with a constant schedule and we select the initial $\lambda$ by monitoring validation performance for each task.

# 5 Experiments and Results

We design and conduct a set of experiments to help answer two main research questions:

- **Q1:** Can we outperform APE and In-Context Learning (ICL) with a DLN-1?
- **Q2:** Does network depth provide further improvement upon DLN-1?

## 5.1 Experimental Setup

**Datasets and Tasks** We adopt a set of nine NLP and reasoning tasks commonly used in prior work studying zero- or few-shot learning capabilities of LLMs [23, 10, 39, 42, 1]. We focus on classification tasks. For tasks adopted from BigBench-Hard (BBH) [42] (Hyper., Nav., Date. and Logic.7[2]), we use the 250 data points provided by BBH as test set. We take the remaining data points from BigBench [39] that were not included in BBH, and randomly split them (evenly) into training and validation sets. For tasks adopted from [23] (Mpqa, Trec, and Subj), we randomly sample 400 and 250 data points from their training and test sets, respectively. We use the original validation sets. For tasks adopted from Leopard [1] (Disaster and Airline), we randomly sample 400, 250, and 250 data points as training, valid, and test. We list all tasks and their statistics in Table 3 in the Appendix.

We use accuracy as the evaluation metric. Specifically, given an input, we compare a system's output string against the ground-truth output string provided by the dataset. We score 1 if the two strings are identical and 0 otherwise. Before the comparison, we process the strings from both the model output and the ground-truth to deal with issues like tokenization and capitalization. In all our DLN experiments, we perform a hyperparameter search and run the same hyperparameter setting with three random seeds. We report the test accuracy averaged over three seeds corresponding to the hyperparameter setting that achieves the highest average validation accuracy. We report details of the hyperparameter search in the Appendix I.

Throughout this paper, we use OpenAI's models, specifically GPT-3 (text-davinci-003) and GPT-4, as the backbone to our proposed systems unless otherwise specified. For DLNs, we use a batch size of 20 and train for 20 iterations by early-stopping on validation performance evaluated every 2 iterations. We then report test scores. We sample $N = 20$ prompt proposals and $K = 5$ hidden samples.

**Baselines** We compare the DLN against two classes of baseline systems. First, we test a set of systems equipped with the same backbone (i.e., GPT-3):

- 0-shot: Given an input, the LLM is required to generate the answer in a zero-shot manner.
- 5-shot (ICL): Given an input as well as five data points as in-context examples, the LLM is queried to generate an answer. The five examples are randomly sampled from the training set.
- KATE [20]: Given an input, we retrieve the five most similar data points from the training set using an off-the-shelf sentence encoder, and use them as in-context examples.
- APE [57]: The LLM is queried to generate a pool of candidate prompts for the task given few input-output pair examples. The candidate prompts are evaluated on a validation set to find the best performing instruction prompt. The best instruction is then used for 0-shot evaluation. We optimize the prompt over both 15 and 400 examples (APE-15 and APE-400 respectively).
- CoT [48]: Given an input, the LLM is first queried to generate a reasoning path with the prompt ``Let's think step by step''. Then, conditioned on the input and its first output, the LLM is queried to generate an answer. This is the zero-shot version of CoT and is a natural baseline for DLN-2: it performs two LLM calls and can be seen as DLN-2 without optimization. We will report performance of this baseline when comparing to DLN-2.

Additionally, we compare against one of the most advanced LLMs to date, GPT-4. We test 0-shot and ICL settings with GPT-4.

## 5.2 DLN-1

Our first set of experiments evaluates the 1-layer language network (DLN-1) described in Section 2. Table 1 presents results on the full suite of test tasks. We see that it matches the performance of

---

[2]We only use the variant with seven objects.

Table 1: Test accuracy averaged over three random seeds of a shallow, 1-layer language network (DLN-1) compared to baselines both on GPT-3 and GPT-4. For trainable systems (i.e., APE and DLN-1) or systems relying on GPT-4, we report the 95% confidence interval.

| Method | BigBench Hard | | | | NLU | | | Leopard | |
|---|---|---|---|---|---|---|---|---|---|
| | Hyper. | Nav. | Date. | Logic.7 | Mpqa | Trec | Subj | Disaster | Airline |
| **GPT-3** | | | | | | | | | |
| 0-shot | 60.8 | 64.1 | 56.4 | 45.9 | 88.0 | 61.9 | 61.7 | 81.6 | 75.6 |
| 5-shot | 55.6 | 56.5 | 62.1 | 36.7 | 87.2 | 80.0 | 76.4 | 81.2 | 82.7 |
| KATE | 71.1 | 56.9 | 61.1 | 44.4 | 88.4 | 77.6 | 71.1 | 76.0 | 81.6 |
| APE-15 | 68.5±5.5 | 67.3±7.7 | 32.1±28.6 | 45.5±4.7 | 85.5±4.6 | 71.3±5.5 | 61.3±7.2 | 54.8±14.6 | 83.5±3.5 |
| APE-400 | 65.5±4.7 | 56.9±32.9 | 23.5±14.1 | 45.6±12.4 | 84.9±9.7 | 72.0±1.7 | 63.7±9.2 | 60.3±37.4 | 82.3±10.0 |
| DLN-1 | 91.9±3.0 | 68.5±4.7 | 55.7±4.5 | 47.5±2.1 | 88.5±2.5 | 89.7±3.2 | 83.2±6.5 | 81.7±6.5 | 83.2±5.5 |
| **GPT-4** | | | | | | | | | |
| 0-shot | 64.0±1.0 | 74.0±1.0 | 79.2±2.6 | 68.5±3.5 | 86.3±0.6 | 64.8±1.7 | 72.5±1.5 | 47.7±0.6 | 84.5±0.6 |
| 5-shot | 88.4±2.6 | 75.7±1.5 | 79.3±1.1 | 62.8±1.7 | 88.0±3.0 | 82.5±3.8 | 94.7±3.5 | 63.6±8.5 | 88.0±1.0 |
| 16-shot | 93.3±2.3 | 75.5±5.1 | 80.9±5.0 | 66.4±3.6 | 91.3±1.5 | 83.7±0.6 | 96.5±2.5 | 67.1±4.0 | 88.3±2.1 |
| DLN-1 | 95.2±5.0 | 77.1±4.7 | 76.7±3.0 | 69.1±2.5 | 91.1±3.2 | 89.5±2.1 | 93.1±5.0 | 82.1±3.8 | 85.9±1.5 |

---

**DLN-1 prompt on Hyperbaton (GPT-3)**

When constructing a sentence with multiple adjectives, the order should be opinion, size, age, shape, color, origin, material, and purpose. Adjectives of the same type should be listed in descending order from largest to smallest. When adjectives of different types are used, the order should be opinion, size, age, shape, color, origin, material, and purpose. For example, in the phrase "massive ancient chair" size (massive) should come before age (ancient). Examples: little old-fashioned Russian silver rectangular ship; silly large old leather hiking chair; brand-new spherical Mexican sweater; enormous old spherical green Nigerian exercise car; medium-size triangular wool eating ship; good square brown Egyptian ship; lovely massive drinking monkey; archaic circular white plastic shoe. In each of the following examples, the adjective order is wrong. Identify the correct adjective order:

---

**DLN-1 prompt on Hyperbaton (GPT-4)**

To determine the correct adjective order, follow this sequence: opinion, size, shape, age, color, origin, material, and purpose. For example, choose "large red plastic ball" over "red large plastic ball" since it follows the order: size (large), color (red), and material (plastic). Not all adjectives may be present, but the order should still be maintained. If the options are "ancient prismlike white leather whittling match" and "leather white ancient prismlike whittling match", choose the first option, as it follows the order: age (ancient), shape (prismlike), color (white), material (leather), and purpose (whittling). Remember that opinion always comes before age, so "obnoxious old-fashioned typing shoe" is correct over "old-fashioned obnoxious typing shoe." Ensure opinion adjectives come before other adjectives in the sequence. When comparing options, follow the order of adjectives for each category: size before color, color before origin, and so on. In cases where purpose and material adjectives are switched, like "paper walking monkey" vs "walking paper monkey", choose the option where material comes before the purpose. Additionally, always prioritize the given sequence over the position of adjectives in the sentences. For example, choose "midsize brand-new gray Chinese wood sweater" over "Chinese brand-new gray midsize wood sweater" as it follows the order: size (midsize), age (brand-new), color (gray), origin (Chinese), and material (wood).

Figure 2: The final prompt of the DLN-1 on Hyperbaton includes not only instructions but also examples from the training set. These samples were automatically chosen by the prompt optimization. In a way, this approach combines in-context learning and prompt optimization.

the best GPT-3-based method on Disaster, Mpqa and Airline and narrowly beats the best GPT-3 baseline on Logic.7 and Nav.. On Hyper., Trec, and Subj, DLN-1 significantly outperforms the best GPT-3 baseline (by about 20, 10, and 7 percentage points, respectively). On Hyper., Trec, and Disaster, it even surpasses GPT-4 baselines, unsurprisingly underperforming GPT-4 on all other tasks. DLN-1's excellent performance on Hyper., a BBH task about ordering adjectives according to linguistic convention, is a surprise. To better understand this result, we show the final prompt in Figure 2. We see that the prompt contains both instructions and a list of examples from the training set. These examples were automatically chosen by the optimizer based on their impact on the performance. This can be seen as a combination of KATE, which selects training examples to put in context based on their similarity with the test example, and APE, which selects the prompt based on its performance. On Date., DLN-1 tends to systematically under-perform the 0-shot baseline both for GPT-3 and GPT-4. We observed that DLN-1 overfits due to paucity of examples in the validation set.

Table 2: DLN-2 test accuracy using GPT-3 as LLM.

| Method | Nav. | Date. | Logic.7 | Disaster | Subj |
|--------|------|-------|---------|----------|------|
| 0-shot | 64.1 | 56.4 | 45.9 | 81.6 | 61.7 |
| CoT | 69.3 | 72.4 | 41.1 | 54.4 | 59.3 |
| APE | $67.3_{\pm7.7}$ | $32.1_{\pm28.5}$ | $45.5_{\pm4.7}$ | $54.8_{\pm14.6}$ | $61.3_{\pm7.2}$ |
| APE-400 | $56.9_{\pm32.9}$ | $23.5_{\pm14.1}$ | $45.6_{\pm12.4}$ | $60.3_{\pm37.4}$ | $63.7_{\pm9.2}$ |
| DLN-1 | $68.5_{\pm4.7}$ | $55.7_{\pm4.5}$ | $\underline{47.5}_{\pm2.1}$ | $81.7_{\pm6.5}$ | $83.2_{\pm5.5}$ |
| DLN-2 | $\underline{83.1}_{\pm24.7}$ | $\underline{75.2}_{\pm14.8}$ | $45.7_{\pm3.5}$ | $\underline{82.8}_{\pm2.5}$ | $\underline{85.9}_{\pm8.7}$ |

## 5.3 DLN-2

We investigate the effectiveness of depth through experiments with 2-layer language networks (DLN-2) on tasks where we expect depth to be most useful, and on which DLN-1 significantly underperforms the GPT-4 0-shot baseline, i.e., Nav., Date., and Logic.7 [42]. Since the Nav., Date. and Logic.7 tasks from BBH require more complex spatial and temporal reasoning, they are the ones where we most expect a decomposition into subtasks to be helpful. We also include Subj and Disaster as an example where DLN-1 performs well (even outperforming the GPT-4 0-shot baseline), since we are interested to see to what extent DLN-2 can further push performance.

Results for DLN-2 can be found in Table 2. Compared to DLN-1, DLN-2 provides an average boost of 7.2% absolute score. On Nav. and Date., DLN-2 largely improves the performance of DLN-1, outperforming all single layer networks. On Logic.7, all methods appear to perform similarly. This could point to the fact that the task might be too hard for the base LLM and thus highlights the limits of prompt optimization of a weak base model. On Subj and Disaster, DLN-2 achieves further improvement over DLN-1. Compared to 0-shot GPT-4 results in Table 1, on Subj and Disaster, DLN-2 on average provides more than 20% in absolute improvement. We encourage readers to find additional experimental results in Appendix C.

## 6 Related Work

**Prompt-Based Machine Learning** GPT-3 [5] launched a new paradigm in NLP called in-context learning (ICL), now applied beyond traditional NLP tasks [21]. The discovery of chain-of-thought prompts (CoT) marked a major advance in prompting: LLM performance improves markedly when the prompt includes examples of intermediate reasoning steps [48] (few-shot CoT), or simply instructs the model to "think step by step" [17] (zero-shot CoT). Like CoT, DLNs break a problem down into intermediate steps but they operationalize these steps as separate LLM calls, each defined by its own learned prompt. Since the introduction of CoT, prompting techniques have evolved to be more dynamic and iterative. Recent methods often operate recursively. Examples include RECITE [41], Self-ask [33], and related methods for question-answering Creswell et al. [6], Zhou et al. [56]. A similar class of methods relies on "introspection" [14], where an LLM is prompted to ingest, evaluate then possibly act on its own previous output. Self-critique [46], ReAct [54], Reflexion [38], Self-refine [24] fit this mould along with Hao et al. [11], Du et al. [9], Yao et al. [53].

**Prompt Optimization** Techniques based on notions of self-talk and self-evaluation align naturally with automatic prompt optimization—a core function in DLNs. Early work in this category includes Autoprompt [37] and GRIPS [32]. Deng et al. [7] argue that 'enumeration-then-selection' heuristics for optimizing discrete prompts do not explore the prompt space systematically. They take an RL approach to overcome this problem, training a policy network, via soft Q-learning with a suitably designed and stabilized reward function, to generate effective prompts. Through Gibbs sampling, Reprompting [52] iteratively searches CoT recipes to improve prompt performance automatically. Most relevant to DLNs, Zhou et al. [57] present Automatic prompt engineer (APE). APE optimizes an initial prompt by searching over a pool of candidates to maximize a score function. We use an APE-inspired approach in DLNs and we cast the proposal/scoring functions as elements of variational inference. In a concurrent work, Pryzant et al. [34] proposed using textual gradients in automatic prompt optimization. This algorithm uses LLM's nonparametric feedback to guide prompt generation and selection.

**Multi-Layer LLM systems** Several recent works compose LLMs as nodes in a computational graph, which is the core idea of DLNs. Some work cited above can be seen as instances of this idea. Similarly, Khot et al. [15] induce an LLM to generate a basic "control flow" that calls distinct LLM modules. Wu et al. [50] propose AI chains, an interactive system of chained LLMs based on a set of "LLM primitive" operations. They conduct a 20-person user study in which participants modify chains, and find this process to improve task performance, transparency, and controllability. Dohan et al. [8] unify LLMs and graphical models as "language model cascades". Specifically, they cast LLM compositions as graphical models with string-valued random variables.[3] They show how scratchpad [30], chain-of-thought [48], tool use [25], and several other prompting strategies fit their formalism. DLNs can likewise be considered an instance of language model cascade, because of that framework's generality. However, going beyond the conceptual work of Dohan et al. [8], we present an effective technique for doing inference in an LLM-based graphical model and we apply learned networks of LLMs to several downstream tasks.

# 7 Conclusion and Future Work

In this paper we introduced an algorithm for joint prompt optimization in deep networks where each layer is an LLM. To do so, we consider outputs of each hidden LLM layer as a latent variable we need to do inference over. From a conceptual perspective, we demonstrated how CoT can be seen as a DLN-2 with a residual connection. Similarly, Generated Knowledge Prompting [19] could be considered as a fixed forward-only DLN-2 where, in the first layer, an LLM generates related knowledge, and in the second layer, another LLM takes the generated knowledge as input and generates the final answer. Other prompting techniques like ReAct [54], Reflexicon [38], and Self-Consistency [46] could all be ensembles of DLN-1s with different prompt initializations.

Although we only tested 1-layer and 2-layer LNs so far, we already show that the performance of smaller LLMs can be boosted when stacked and prompted properly. We believe the modularity of these architectures will make them more adaptable and reusable to new use cases. While accuracy on downstream tasks is an appealing metric, we argue that other considerations are just as important, for example the ease of adapting a model to one's own use case, or the ability to leverage multiple existing models.

We noticed that GPT-3 has a tendency to always produce an answer given an example: this could be due to the particular 0-shot fine-tuning procedure, which biases the model towards generating useful responses. This raises the question of whether we can fine-tune "stackable" LLMs and whether DLNs can be used as a framework to generate training data for that purpose. Second, we engineered our backward and forward templates; in the future, we wish to expand our work to learn parts of such templates: we expect this to make the variational bound tighter and thus easing DLN's optimization. Additionally, while we only proposed 2-layer DLNs, the framework accommodates arbitrary directed acyclic graphs.

**Impact statement** While we are fully aware of the limitations of addressing societal issues through technical work, we hope that modular approaches like ours will alleviate some of the issues associated with LLMs, like the concentration of power associated with the difficulty to train them. We also hope that, by facilitating the reusability and adaptivity of such models, we shall make them more amenable to a wider variety of use cases. However, while we discuss the performance of these models on artificial benchmarks, we do not address the question of when and how such models should be deployed, nor do we offer additional guarantees against their misuse. We also emphasize that performance on artificial tasks, even if realistic, is neither representative of performance in uncontrolled environments, nor enough to justify the deployment of these models in high stakes situations.

**Acknowledgements** We would like to acknowledge Silviu Pitis for the useful feedback on the draft, Nikolay Malkin and Tong Wang for their advice during the first steps of this project.

---

[3] Earlier work by Miao and Blunsom [27] also treated strings as random variables.

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

## Contents in Appendices:

- In Appendix A, we list the contribution of each author to this work.
- In Appendix B, we provide additional experimental details including task statistics and the prompt strings we used to initialize DLN.
- In Appendix C, we provide additional experiments and baselines we compare to.
- In Appendix D, we provide forward and backward templates being used in DLN.
- In Appendix E, we provide an algorithm that generalizes DLN training in a multiple layer setting.
- In Appendix F, we show examples of learned weights that exhibit behavior similar to in-context learning.
- In Appendix G, we show examples of learned weights by 2-Layer DLNs.
- In Appendix H, we show an example of the hidden states produced by a 2-Layer DLN.
- In Appendix I, we provide implementation details, including hyperparameter information.
- In Appendix J, we discuss resource used in DLN development and their pricing.

Table 3: Tasks used in this work.

| Task | \|train\| | \|valid\| | \|test\| | \|class\| | Description |
|------|------|------|------|-------|-------------|
| Mpqa | 400 | 256 | 250 | 2 | Sentiment analysis. |
| Trec | 400 | 256 | 250 | 6 | Question type classification. |
| Subj | 400 | 256 | 250 | 2 | Determine whether a sentence is subjective or objective. |
| Disaster | 400 | 250 | 250 | 2 | Determine whether a sentence is relevant to a disaster. |
| Airline | 400 | 250 | 250 | 3 | Airline tweet sentiment analysis. |
| Hyper. | 400 | 1000 | 250 | 2 | Order adjectives correctly in English sentences. |
| Nav. | 375 | 375 | 250 | 2 | Spatial reasoning given navigation instructions. |
| Date. | 59 | 60 | 250 | 6 | Infer a date from context. |
| Logic.7 | 225 | 225 | 250 | 7 | Deduce the order of seven objects given instruction. |

# A    Contributions

Alessandro Sordoni proposed the general idea of DLN, where multiple prompts are learnt at each layer through backward natural language operations; they proposed to generate synthetic in-context examples and the exploration reward for DLN-2; they wrote the code and ran the experiments; they focused on Sections 2, 3, 4 and contribute writing the rest of the sections.

Xingdi Yuan co-developed the basic idea of DLN and wrote part of the code, they also co-designed and helped conducting experiments. They contributed to the writing of the paper, mainly Sections 5 and 6.

Marc-Alexandre Côté helped with the experiments and the infrastructure to make calls to OpenAI models. They also built a demo to visualize the evolution of DLN's prompts during training and contributed to the writing of the paper, mainly focusing on the algorithms and the appendix.

Matheus Pereira co-coded an earlier, non-variational backwards operator with AT, helped with the APE and DLN-2 layers experiments, implemented the method for estimating the total cost of experiments, build a demo to visualize the evolution of DLN's prompts during training, and contributed to the release of the DLN code.

Adam Trischler helped with template conception and iteration, co-coded an earlier, non-variational backwards operator with MP, and contributed to paper writing, mainly the literature review.

Ziang Xiao helped with the model evaluation and experiment setup and contributed to the paper writing, mainly the literature review and discussion.

Arian Hosseini participated in the development discussions throughout the project and contributed to writing the literature review of the paper.

Friederike Niedtner organized and managed the project, helping the team focus on the right priorities.

Nicolas Le Roux proposed the variational inference formulation and the posterior sharpening. They offered guidance and mentorship for the project. They also contributed to the writing of the paper, mainly sections 1, 2, and 3.

# B    Additional Experimental Details

## B.1    Additional Task Information

In Table 3, we provide short descriptions for all tasks we use and their statistics.

## B.2    Prompt Initialization

We initialize the "classification" layer of the DLNs, i.e. the first layer of the 1-Layer LN and the second layer of the 2-layer DLN, with a question or a task description as reported in Table 4. We use these same initializations to compute the 0-shot performance. Therefore, at initialization, a 1-layer LN is equivalent to the 0-shot baseline. For the hidden layer of the 2-layer DLN, we initialize the prompt to ``Decompose the problem to make it simpler:'' for Nav. and Subj, and ``''

Table 4: Prompt initializations.

| Task | Initialization |
|---|---|
| Mpqa | Read the following review, then choose whether it is negative or positive. |
| Trec | Read the following question, then choose whether it is about a description, entity, expression, human, location or number. |
| Subj | Read the following sentence, then choose whether it is subjective or objective |
| Disaster | Read the following sentence, then choose whether it is relevant to a disaster. |
| Airline | Read the following sentence, then choose whether it is positive, negative, or neutral. |
| Hyper. | Which sentence has the correct adjective order. |
| Nav. | Read the following sentence, then determine whether you return to the starting point. |
| Date. | Infer the date from context. |
| Logic.7 | The following paragraphs each describe a set of seven objects arranged in a fixed order. The statements are logically consistent within each paragraph. |

Table 5: Test accuracy averaged over three random seeds. We compare the layerwise and the end-to-end trainings for DLN-2.

| | Nav. | Date. | Logic.7 | Subj |
|---|---|---|---|---|
| DLN-2 *(fix 2nd)* | 73.1 | 61.6 | 43.3 | 80.2 |
| DLN-2 *(fine-tune 2nd)* | 76.4 | 62.8 | 40.7 | 84.5 |
| DLN-2 *(end-to-end)* | 83.1 | 75.2 | 45.7 | 85.9 |

(empty string) for Date. and Logic.7. We didn't try other initializations for this hidden layer, we leave this for future explorations.

## C   Additional Experiments

In addition to the prompt engineering and few-shot baselines from the main paper, we include here comparisons to more of those on a subset of the datasets. Particularly we added results for:

- Table 6 shows DLN-1 and DLN-2 outperforming CoT+APE (implemented as described in Section 4.3 of Zhou et al. [57]);
- Table 7 shows that even increasing the number of examples for ICL and KATE cannot match DLN-1 and DLN-2 performance;
- Table 8 and Table 9 show results for DLN-1 using open-source language models such as WizardLM-13b-v1.2 [51] and LLaMA2-70B-Chat [45] respectively.

**Layerwise training of DLN-2**    We also explored a different learning strategy for DLN-2: layer-wise pre-training. We start the learning of a single layer DLN using the technique from Section 2. We call the prompt obtained at the end of this optimization $\pi^*$. Then, we train a DLN-2 by initializing $\pi_1 = \pi^*$, and then learn $\pi_0$, the parameters of the bottom layer, using variational inference. We explore two variants: one keeps the last layer fixed and one it fine-tunes the last layer. We report their results in Table 5.

Table 6: Test accuracy averaged over three random seeds with 95% confidence interval. All models use GPT-3. DLN-1 and DLN-2 outperform CoT+APE.

| Method | Hyper. | Nav. | Date. | Logic.7 |
|---|---|---|---|---|
| APE-15 | $68.5_{\pm 5.5}$ | $67.3_{\pm 7.7}$ | $32.1_{\pm 28.6}$ | $45.5_{\pm 4.7}$ |
| CoT+APE | $50.9_{\pm 0.8}$ | $61.5_{\pm 1.5}$ | $58.6_{\pm 2.2}$ | $38.9_{\pm 1.6}$ |
| DLN-1 | $\underline{91.9}_{\pm 3.0}$ | $68.5_{\pm 4.7}$ | $55.7_{\pm 4.5}$ | $\underline{47.5}_{\pm 2.1}$ |
| DLN-2 | - | $\underline{83.1}_{\pm 24.7}$ | $\underline{75.2}_{\pm 14.8}$ | $45.7_{\pm 3.5}$ |

Table 7: Test accuracy averaged over three random seeds with 95% confidence interval (where applicable). All models use GPT-3. Increasing the number of ICL examples helps performance, but cannot match DLN-1 and DLN-2 in general. Context length limit is an issue for ICL and KATE 32-shot.

| Method | Nav. | Date. | Logic.7 | Subj |
|---|---|---|---|---|
| ICL - 5-shot | 56.5 | 62.1 | 36.7 | 76.4 |
| ICL - 10-shot | 61.3 | 62.9 | 38.9 | 72.0 |
| ICL - 32-shot | 66.0 | 63.5 | - | 83.2 |
| KATE - 5-shot | 56.9 | 61.1 | 44.4 | 71.1 |
| KATE - 10-shot | 59.5 | 62.0 | 41.6 | 73.9 |
| KATE - 32-shot | 67.5 | 62.8 | - | 80.4 |
| DLN-1 | $68.5_{\pm 4.7}$ | $55.7_{\pm 4.5}$ | $\underline{47.5}_{\pm 2.1}$ | $83.2_{\pm 5.5}$ |
| DLN-2 | $\underline{83.1}_{\pm 24.7}$ | $\underline{75.2}_{\pm 14.8}$ | $45.7_{\pm 3.5}$ | $\underline{85.9}_{\pm 8.7}$ |

Table 8: Test accuracy using WizardLM-v1.2 13B as LLM. This open source model seems significantly less able to capture few-shot examples from the context. DLN-1 outperforms ICL on all tasks here.

| Method | Nav. | Logic.7 | Subj |
|---|---|---|---|
| 0-shot | 58.0 | 0.0 | 65.8 |
| 5-shot | 56.0 | 28.0 | 50.8 |
| DLN-1 | $\underline{61.1}$ | $\underline{31.0}$ | $\underline{79.8}$ |

Table 9: Test accuracy averaged over three random seeds with 95% confidence interval. All methods use LLaMA2-70B-Chat as LLM, with DLN + GPT3 employing text-davinci-003 as the backward LLM for prompt and hidden proposals.

| Method | Nav. | Date. | Logic.7 | Subj |
|---|---|---|---|---|
| 0-shot | $42.0_{\pm 0.0}$ | $25.2_{\pm 0.0}$ | $14.4_{\pm 0.0}$ | $62.4_{\pm 0.0}$ |
| 5-shot | $43.2_{\pm 12.5}$ | $21.1_{\pm 9.7}$ | $16.4_{\pm 2.6}$ | $67.7_{\pm 15.5}$ |
| DLN-1 + GPT3 | $43.6_{\pm 4.0}$ | $21.9_{\pm 5.7}$ | $33.1_{\pm 10.9}$ | $\underline{80.9}_{\pm 11.5}$ |
| DLN-1 | $44.9_{\pm 6.4}$ | $31.6_{\pm 10.5}$ | $\underline{38.4}_{\pm 3.6}$ | $76.1_{\pm 4.5}$ |
| DLN-2 + GPT3 | $43.7_{\pm 3.0}$ | $51.1_{\pm 4.0}$ | $21.9_{\pm 4.9}$ | $59.1_{\pm 16.7}$ |
| DLN-2 | $\underline{68.9}_{\pm 14.4}$ | $\underline{61.7}_{\pm 17.6}$ | $20.0_{\pm 13.7}$ | $63.1_{\pm 25.4}$ |

## D Templates

### D.1 Forward Templates

**"Classification" Template** F **for 1-Layer LN** In the 1-layer LN, we use the following template to elicit the output $y$ given the input $x$. prompt is substituted with the value of the current prompt.

> **Classification Template** F
>
> **template**:
>    {{ **prompt** }}
>
>    {{ **input** }}
>
>    Answer:

**Residual Classification Template** $F_r$ **for 2-Layer DLN** In the 2-layer LN, the last layer just concatenates the input to the output of the first layer, $\hat{h}$, before eliciting an answer.

> **Residual classification template** $F_r$
>
> **template**:
>    {{ **prompt** }}
>
>    {{ **input** }}
>    Your thoughts were:
>    {{ **h** }}
>
>    Answer:

**Hidden Layer** F The variable prompt is substituted with the value of the current prompt $\pi_0$. This has the effect of providing additional information about how "Let's think step by step" should behave.

> **Hidden Layer** F
>
> **template**:
>    {{ **input** }}
>
>    {{ **prompt** }} Let's think step by step.

For 2-Layer DLN on Subj, we use the following hidden template. We couldn't run with the previous template due to lack of time, as we observed that the step by step trigger tended to generate lengthy hidden states.

> **Hidden Layer** F
>
> **template**:
>    {{ **prompt** }}
>
>    {{ **input** }}
>
>    Brief Analysis:

### D.2 Backward Templates (Prompt and Hidden Proposals)

**Prompt Proposal Template** $B_\pi$ $B_\pi$ is used to propose new candidate prompts. The template takes as input the current prompt, prompt, and a mini-batch of examples stored in backward_infos. backward_info.input stores the input to the layer ($x$ if we are proposing prompts for the first layer or $h$ if it is the second layer); backward_info.target stores the target to the layer ($h^*$ if it is the first layer or $y$ otherwise); backward_info.output stores the predictions of the model during the forward pass ($\hat{h}$ if it is the first layer, and $\hat{y}$ if it is the second layer). message is substituted with one of the message_alternatives, sampled at random during the DLN training. This induces diversity in the generated prompts and allows emergence of learning to ICL behaviors, where the prompts contain synthetic examples that can help solve the task.

**Backward Hidden Templates** $B_h$ We experiment with multiple backward templates to sample hidden states from the approximate posterior distribution $q(h)$. Each vary in the amount of conditioning information. The simpler way of sampling hidden states is to use the same hidden template as in the forward pass. This corresponds to using a posterior distribution $q(h)$ which is equivalent to the prior distribution $p(h)$. The other alternative is to condition the posterior template on the answer $y$, as illustrated below:

The next alternative we experiment with is a more verbose template that takes as input the prompt for the final layer $\pi_1$ in `next_prompt`, the input $x$ in `input`, the hidden state from the forward pass $\hat{h}$ in `h` and the ground-truth output $y$. We use a similar strategy of sampling different message alternatives to substitute with `message` to increase diversity of the hidden samples:

In practice, we found that sampling from hidden states from a mixture of forward template F, i.e. $p_{\texttt{LM}}(h|x,\pi_0)$, and backward template with $y$ conditioning, i.e. $q(h|x,y,\pi_0)$ works well. We suspect that this has the effect of capping the KL divergence term between posterior and prior distribution, i.e. the $\text{KL}(q(h)||p_{\texttt{LM}}(h|x,\pi_0))$ appearing in the ELBO. In the future, this could be addressed in a more principled way by learning a prompt for the posterior proposal.

# E   Generalized VI to Multiple Layers

We report the generalized training algorithm for multiple layers in Algorithm 3.

# F   Learning to In-Context Learn: Additional Examples

In Figure 2, we report examples of prompts found by the 1-layer DLN on the Hyperbaton task, which exhibit either integration or verbalization of task examples. Here, we provide additional examples of prompts found by DLN-1 on the MPQA task.

---

**DLN-1 prompt on MPQA (GPT-3)**

Read each sentence, then decide if the sentence is expressing a positive or negative sentiment. For example, if the sentence is "supported", choose "positive", and if the sentence is "derail", choose "negative". Similarly, if the sentence is "victorious", choose "positive", and if the sentence is "would not be a bad idea", choose "positive". Additionally, if the sentence contains multiple words, consider the overall sentiment of the sentence and choose the appropriate option. For example, if the sentence is "counting on", choose "negative", and if the sentence is "peace and stability and prosperity", choose "positive". Note that words like "artificial" tend to have a negative sentiment.

---

**DLN-1 prompt on MPQA (GPT-4)**

Determine whether the given input has a positive or negative connotation by analyzing the meaning of the words and phrases in context. If the input expresses a favorable, desirable, or pleasant meaning, choose "positive." If the input expresses an unfavorable, undesirable, or unpleasant meaning, choose "negative." Consider the overall sentiment expressed by the input rather than focusing on individual words or phrases. For example, "calling for" generally has a positive connotation as it implies advocating or supporting something, while "a true Muslim fighter" can be seen as positive, since it refers to someone dedicated to their beliefs. Keep in mind that some phrases may have a positive connotation when they imply improvement or resolution, like "put an end to." Additionally, phrases like "to the contrary" can have a positive connotation when they suggest a differing, yet valid perspective or opinion. When analyzing the input, consider the context in which it is used, as the connotation of a word or phrase can change depending on the situation. For example, "extra vigil" can have a positive connotation when it implies increased awareness and preparedness.

---

# G    Examples of 2-Layer Best Weights (GPT-3)

## G.1    Navigate (81.6% Dev Acc)

> **DLN-2 Prompt:** $\pi_0$
>
> Decompose the problem by breaking it down into steps and describing the new position after each step. Note that the direction you are facing stays the same unless specified. Start by specifying the direction you are facing and the coordinates of the starting point. For each step, describe the number of steps taken and the direction of movement (e.g. North, South, East, or West). Specify the new coordinates and direction after each step.

> **DLN-2 Prompt:** $\pi_1$
>
> Start facing north. Take the specified number of steps in the indicated direction and turn when specified. Make sure to keep track of your direction and the number of steps taken to ensure you return to the starting point.

## G.2    Subj (89.9% Dev Acc)

> **DLN-2 Prompt:** $\pi_0$
>
> Reflect on the input to produce an output that is either objective (factual) or subjective (involving opinion or value judgment). Objective statements describe events or facts that can be verified, while subjective statements express opinion or personal feelings about a fact, event, or situation that cannot be confirmed as an accurate statement of fact. For example, if the input is "the film was a success at the box office," the output would be "This statement is an objective fact, as it describes events that can be verified." If the input is "the film was an amazing experience," the output would be "This statement is subjective, expressing a personal opinion about the film in question that reflects approval and judgement, and cannot be confirmed as an accurate statement." If the input is "neither Juan Antonio nor Señor Maximiliano know what they are in for when the tables are turned by the sly Carmen," the output would be "This statement is an objective fact, as it describes events that can be verified regarding the actions of Juan Antonio, Señor, and Maximiliano, and the change of circumstances caused by Carmen." If the input is "humorless, self-conscious art drivel, made without a glimmer of intelligence or invention," the output would be "This statement is subjective, expressing a negative opinion about the film in question that reflects disapproval and judgement, and cannot be confirmed as an accurate statement." If the input is a hypothetical situation, such as "how would you feel if when you woke, the nightmare had just begun?", the output would be "This statement is subjective, expressing a personal opinion about a hypothetical situation that cannot be verified."

> **DLN-2 Prompt:** $\pi_1$
>
> Read the sentence. Determine if it is expressing a fact or opinion. A fact is an accurate statement that can be confirmed, while an opinion is a personal viewpoint that reflects someone's beliefs. Facts are typically statements about something that happened, such as events, actions, or conditions, or statements that describe a state of being, such as someone's personality or a physical object. Opinions are typically statements that express judgement, approval, or disapproval. Examples of facts include statements about events that occurred, such as "the film was released in 2012," or statements about conditions, such as "the weather is sunny," or statements that describe a state of being, such as "the detective is strong and independent." Examples of opinions include statements about how good or bad something is, such as "the film was terrible" or "the weather is beautiful." Examples of questions that are not facts or opinions include hypothetical questions, such as "how would you feel if when you woke, the nightmare had just begun?". Select "objective" for a fact and "subjective" for an opinion.

# H    Examples of Hidden States

We report the forward pass produced by a 2-Layer DLN in Section G.1 on Navigate below.

> **Input** $x$
>
> If you follow these instructions, do you return to the starting point? Take 3 steps. Take 10 steps. Take 4 steps. Take 1 step.
> Options:
> - Yes
> - No

> **Hidden** $\hat{h}$
>
> 1. Take 3 steps: (3, 0) facing east
> 2. Take 10 steps: (13, 0) facing east
> 3. Take 4 steps: (17, 0) facing east
> 4. Take 1 step: (18, 0) facing east
>
> Answer: No, you do not return to the starting point.

Table 10: Hyperparameter search space.

| hyperparam | search space |
|---|---|
| **1-Layer LN** | |
| bh_tpl | q_action_prompt:v3.0, q_action_prompt:v3.5 |
| tolerance | -1, 0, 2 |
| use_memory | 0, 2 |
| held_out_prompt_ranking | True, False |
| **2-Layer DLN PT + fix 2nd layer** | |
| bh_tpl | q_action_prompt:v3.0, q_action_prompt:v3.5 |
| logp_penalty | 0., 0.5, 2. |
| **2-Layer DLN PT + fine-tune 2nd layer** | |
| bh_tpl | q_action_prompt:v3.0, q_action_prompt:v3.5 |
| logp_penalty | 0., 0.5, 2. |
| **2-Layer DLN end-to-end** | |
| num_h_samples | 5, 10 |

---

**Output** $\hat{y}$

No

---

# I   Implementation Details

We report hyperparameter search space in Table 10. A brief description of the hyperparameters is as follows:

- `bh_tpl` is the type of backward prompt template we use $B_\pi$. v3.5 is equal to the $B_\pi$ we report in Section D. In v3.0, we remove "Be concise." at the end of each `message_alternatives`. We noticed that in general v3.5 works better as it implements a sort of regularization on the length of the found prompts. Future work could address length regularization in a more principled manner.

- `logp_penalty` is the coefficient for the exploration reward we mentioned in the paper.

- `num_h_samples` is the number of $h$ samples to generate from the approximate posterior distribution.

- `use_memory` is whether or not we use the backtracking mechanism. Usually 2 works well across tasks.

- `held_out_prompt_ranking` describes whether we use only half of the mini-batch examples for each prompt proposal, as described in the main paper.

- `tolerance` describes after how many iterations we reload the best weights found during the last validation if the current validation score is lower than the best score obtained so far.

For the 2-Layer experiments, we have to restrict this search space due to computational costs. We use bh_tpl = "v3.5", tolerance = 2, use_memory = 2, held_out_prompt_ranking = True, logp_penalty = 0.5.

# J   Pricing

We keep track the number of tokens we interact with GPT-3 via its online API. According to OpenAI's pricing policy, user pays for both the input tokens (prompts) and the output tokens. Using the Hyperbaton task as an example, while training a 1-layer LN, the total number of tokens we use is

Table 11: Test accuracy along with inference cost expressed in tokens (in gray) averaged over three random seeds of a shallow, 1-layer language network (DLN-1) compared to baselines on GPT-4. We also report the 95% confidence interval on the test accuracy. We emphasize the cost at the testing time because it is more relevant in real-world deployment and the training cost is one-off.

| Method | BigBench Hard | | | | NLU | | | Leopard | |
|---|---|---|---|---|---|---|---|---|---|
| | Hyper. | Nav. | Date. | Logic.7 | Mpqa | Trec | Subj | Disaster | Airline |
| **GPT-4** | | | | | | | | | |
| 0-shot | 64.0±1.0 | 74.0±1.0 | 79.2±2.6 | 68.5±3.5 | 86.3±0.6 | 64.8±1.7 | 72.5±1.5 | 47.7±0.6 | 84.5±0.6 |
| | (7.6k) | (12.9k) | (23.6k) | (46.2k) | (3.7k) | (7.6k) | (10.2k) | (10.1k) | (9.9k) |
| 5-shot | 88.4±2.6 | 75.7±1.5 | 79.3±1.1 | 62.8±1.7 | 88.0±3.0 | 82.5±3.8 | 94.7±3.5 | 63.6±8.5 | 88.0±1.0 |
| | (48.0k) | (79.2k) | (143.3k) | (287.5k) | (24.5k) | (52.5k) | (62.6k) | (63.5k) | (61.7k) |
| 16-shot | 93.3±2.3 | 75.5±5.1 | 80.9±5.0 | 66.4±3.6 | 91.3±1.5 | 83.7±0.6 | 96.5±2.5 | 67.1±4.0 | 88.3±2.1 |
| | (136.8k) | (229.9k) | (405.1k) | (817.6k) | (70.3k) | (149.0k) | (177.9k) | (179.4k) | (175.2k) |
| DLN-1 | 95.2±5.0 | 77.1±4.7 | 74.3±1.5 | 69.1±2.5 | 91.1±3.2 | 89.5±2.1 | 93.1±5.0 | 82.1±3.8 | 85.9±1.5 |
| | (77.2k) | (29.9k) | (52.3k) | (68.5k) | (65.4k) | (120.7k) | (46.5k) | (47.1k) | (38.2k) |

---

**Algorithm 3** Deep Language Network Training Algorithm

---

1: $\hat{h} \sim p_{\text{LM}}^t(x)$: generates a completion of prefix $x$ with temperature $t$.
2: $\log p_{\text{LM}}(h|x)$: return log-prob of $h$ following $x$.
3: $N$: # prompt samples, $K$: # posterior samples, $I$: # iterations, $L$: # layers, $\mathcal{D}$: dataset
4: F: template for the inference (forward pass).
5: $\text{B}_\pi$, $\text{B}_h$: templates for prompt and hidden proposal (backward pass).
6: Initialize all layers $\pi_l$ with a generic sentence or task description.
7: **for** $i$ in $[1, I]$ **do**
8:      $x, y \sim \mathcal{D}$                          ▷ Sample minibatch
9:      $\hat{h}_0 \leftarrow x$
10:      $\hat{h}_l, \ldots, \hat{h}_L \leftarrow p_{\text{LM}}^0(\text{F}(\hat{h}_{l-1}, \pi_{l-1}))$          ▷ Inference pass for all layers $0 < l \leq L$
11:      $h_L^* \leftarrow y$
12:      **for** $l$ in $[L-1, 1]$ **do**
13:          $h_l^1, \ldots, h_l^K \sim p_{\text{LM}}^{0.7}(\text{B}_h(\hat{h}_l, h_{l+1}^*, \pi_l))$      ▷ Sample $K$ posterior proposals for $h_l$
14:          $\alpha_l^1, \ldots, \alpha_l^K \leftarrow \log p_{\text{LM}}(h_l^k|\text{F}(\hat{h}_{l-1}, \pi_{l-1}))$ ▷ Compute prior log-probs for all $h_l^k$ samples
15:          $\beta_l^1, \ldots, \beta_l^K \leftarrow \log p_{\text{LM}}(h_{l+1}^*|\text{F}(h_l^k, \pi_l))$     ▷ Compute log-likelihoods for all $h_l^k$ samples
16:          $q_l^k \leftarrow \exp(\alpha_l^k + \beta_l^k)/(\sum_k \exp(\alpha_l^k + \beta_l^k))$     ▷ Compute normalized posterior weights
17:          $h_l^* \leftarrow \arg\max_{h_l}\{q_l^1, \ldots, q_l^K\}$          ▷ Select best posterior sample for layer $l$
18:      **end for**
19:      **for** $l$ in $[L-1, 0]$ **do**
20:          $\pi_l^1, \ldots, \pi_l^N \sim p_{\text{LM}}^{0.7}(\text{B}_\pi(\{\hat{h}_l, \hat{h}_{l+1}, h_{l+1}^*\}, \pi_l))$       ▷ Sample $N$ candidate prompts for $\pi_l$
21:          $s_l^1, \ldots, s_l^N \leftarrow \sum_k \sum_{k'} q_l^k q_{l+1}^{k'} \log p_{\text{LM}}(h_{l+1}^{k'}|\text{F}(h_l^k, \pi_l^n))$     ▷ Compute ELBO for all prompts $\pi_l^n$
22:          $\pi_l \leftarrow \arg\max_{\pi_l^n}\{s_l^1, \ldots, s_l^N\}$          ▷ Select prompt $\pi_l$ with best score
23:      **end for**
24: **end for**

---

2,941,360. For a 2-layer DLN, the total number of tokens we use is 13,654,962. According to the current price for GPT-3 ($0.02/1k tokens), a single run of a 1-layer and 2-layer DLN cost roughly 59 USD and 273 USD, respectively.

In Table 11, we report the cost (lower is better) in terms of total number of tokens for the test set (prompts included). We emphasize the cost at the testing time because it is more relevant in real-world deployment and the training cost is one-off. We can see DLN-1 improves over ICL on 5 out of 9 tasks on GPT-4 at a comparable token cost. Some tasks do not benefit from ICL (i.e. reasoning tasks) while other tasks like Subj, Trec, and Hyper. benefit significantly.

