# OpenReview forum: "Joint Prompt Optimization of Stacked LLMs using Variational Inference"
_NeurIPS.cc/2023/Conference — NeurIPS 2023 poster_

### Official Review · Reviewer_Y4LZ · 2023-07-01

**Soundness:** 3 good
**Presentation:** 3 good
**Contribution:** 4 excellent
**Rating:** 7
**Confidence:** 4

**Summary:**

The paper introduces a framework called Deep Language Network (DLN) that involves stacking multiple large language models (LLMs) as stochastic layers in a deep network. The prompts at each layer serve as tunable parameters, and the output of one layer is fed as input to the next layer. The LLMs are trained jointly using variational inference. The DLN architecture achieves higher performance than a single layer and can sometimes match the performance of larger and more powerful models like GPT-4. The authors discuss the analogy between LLMs and traditional parametric and nonparametric models, explore the limitations of LLMs, propose the use of variational inference for training DLN, and demonstrate the performance of DLN on various datasets. They suggest that DLN can serve as a framework for characterizing and optimizing prompt learning techniques. The paper also discusses future directions such as testing with different language models, fine-tuning stackable LLMs, and expanding DLN to accommodate arbitrarily directed acyclic graphs. The authors hope that DLN and modular approaches like theirs will address the challenges associated with LLMs and make them more adaptable to different use cases. However, they acknowledge the limitations of technical solutions and the need to consider deployment and ethical considerations when using such models in real-world applications.

**Strengths:**

The paper introduces the concept of Deep Language Networks (DLNs), which stack multiple Large Language Models (LLMs) and train them jointly using variational inference. This approach offers a new perspective on leveraging the power of LLMs and demonstrates improved performance compared to a single-layer model. It presents a layered architecture that decomposes the task into a series of smaller sub-tasks, each of which is more easily solvable by an LLM. This decomposition allows for more efficient training and better performance. It details prompt engineering and in-context learning techniques for optimizing the prompts associated with each layer of the DLN. These techniques enable fine-tuning of the DLN's performance and allow for better adaptation to different tasks and datasets. The paper utilizes variational inference to learn the prompts in the DLN. This approach enables the joint search over the prompts, addressing the challenge of optimizing multiple prompts in a deep architecture. The paper provides experimental results on various datasets, demonstrating the effectiveness of DLNs compared to single-layer models. The results show that DLNs can achieve performance comparable to higher-capacity models like GPT-4, even when each LLM in the network is smaller and less powerful. It discusses practical aspects of DLN implementation, such as proposal diversity, learning in-context learning, and backtracking and memory optimization. These considerations make the DLN approach more feasible and effective in real-world scenarios. The strengths of the paper lie in its innovative approach, theoretical framework, practical insights, and experimental validation, showcasing the potential of Deep Language Networks for improving language modeling tasks.

**Weaknesses:**

The proposed Deep Language Networks (DLNs) introduce a more complex architecture compared to single-layer models. This complexity may make the implementation and training of DLNs more challenging, requiring significant computational resources and expertise. Although the paper compares DLNs to single-layer models, it does not provide a comprehensive comparison to other state-of-the-art language models or architectures. Without such comparisons, it is difficult to assess the relative performance and advantages of DLNs against other advanced models. Additional evaluation metrics would provide a more comprehensive assessment of DLNs' strengths and weaknesses. The experiments and results presented in the paper may not generalize to all types of language modeling tasks and datasets. The performance of DLNs could vary depending on the specific domain, language, or task requirements. The paper should provide a detailed analysis of the limitations and potential failure cases of DLNs in different scenarios. DLNs consist of multiple layers of Large Language Models (LLMs), which can increase the computational overhead during training and inference. It does not extensively discuss the computational efficiency of DLNs, including the training time, memory requirements, and inference latency. These factors could limit the scalability and applicability of DLNs in resource-constrained environments.

**Questions:**

Can you provide more details on the computational and memory requirements of the proposed Deep Language Networks (DLNs) compared to single-layer models?

Are there any specific challenges or limitations associated with implementing and training DLNs that were not mentioned in the paper?

Could you include a more comprehensive comparison of DLNs against other state-of-the-art language models or architectures? This would help understand the relative performance and advantages of DLNs in a broader context.

Are there any plans to explore alternative metrics in future work?

**Limitations:**

Yes, the content you provided appears to adequately address the requirement of addressing the limitations and potential negative social impact of the authors' work. The authors acknowledge the limitations of addressing societal issues through technical work and emphasize that their modular approach aims to alleviate some of the issues associated with large language models (LLMs), such as concentration of power and difficulty in training. They also express the hope that their approach will make LLMs more adaptable and suitable for a wider range of use cases. However, the authors explicitly state that they do not address the deployment of such models, when and how they should be used, or provide additional guarantees against their misuse. They highlight the importance of considering the performance of these models in uncontrolled environments and the need for justification before deploying them in high-stakes situations. By acknowledging these limitations and potential negative societal impacts, the authors demonstrate transparency and responsibility. They recognize the need for further consideration of ethical and societal implications beyond the technical performance of their work. The content provided covers the requirement and fulfills the guidelines for addressing limitations and broader societal impacts as specified in the checklist.

---

> ### Author Rebuttal · Authors · 2023-08-09
>
> Thanks for your review! We are glad that you appreciated our work.
>
> ---
>
> ***“comparison to other state-of-the-art language models or architectures”***
>
> We provide additional experimental results as listed in the general response. Please refer to ***Table A/B/C/D*** in the general response for these comparisons.
>
> ---
>
> ***“analysis of the limitations and potential failure cases of DLNs in different scenarios”***
>
> We will discuss limitations of DLNs more in depth in the camera ready. Notably, one of the biggest limitation is that the base LLM cannot be *too small*. This can be seen in Logic.7 for example, which is the hardest task in our benchmark, where every layer must do a notable amount of computation. So there is a tradeoff between capacity of the base LLM and depth of the DLN that is just not discussed enough in the paper right now and that we will address in the camera ready.
>
> ---
>
> ***“Computational efficiency, including the training time, memory requirements, and inference latency”***
>
> Memory: We emphasize that DLN does not have neural net components per se, so there is no GPU needed on a local machine. We require access to a endpoint that can be either local or remote. If the endpoint is local, this needs to be powered by local GPU resources of course.
>
> Training time / inference latency: we are strongly dependent on traffic of OpenAI API. Our DLN-2 results roughly take 1 hour to execute (when without too much traffic).
>
> We are working to relax DLN's dependence from online APIs, specifically, we are looking into applying DLN on open sourced LLMs, such as WizardLM. Please refer to ***Table D*** in the general response for some preliminary results.
>
> ---
>
> ***“Alternative metrics”***
>
> To our understanding, the reviewer refers to the scoring functions DLN uses to obtain training signals. Indeed, assuming access to output log probabilities somewhat makes DLN more difficult to use blackbox LLMs (e.g., GPT-4) as backbone. We agree alternative training signals could be a useful future direction. In fact, we are actively exploring alternative ways to help DLNs learn. In our additional experimental results (***Table C***), we include a variant of DLN-1 using GPT-4 as backbone. Specifically, we use accuracy as the final scoring function in that setting. Devising a scoring function for the hidden layer in the case where the log-probabilities are not available (i.e. for GPT-4) is a future direction.

---

> > ### Comment · Reviewer_Y4LZ · 2023-08-17
> > **Looking forward to**
> >
> > Thanks for the clarifications, looking forwarding the seeing your updates.

---

### Official Review · Reviewer_SaT7 · 2023-07-05

**Soundness:** 2 fair
**Presentation:** 2 fair
**Contribution:** 2 fair
**Rating:** 4
**Confidence:** 4

**Summary:**

This paper suggests stacking multiple large language models (LLMs) together, with tunable parameters represented as prompts at each layer. Given that these prompts are discrete natural language elements, their direct optimization using a gradient-based method is challenging. To address this, the authors propose a prompt optimization framework: (1) generating N local candidates using a proposal distribution, and (2) scoring each candidate to select the one that maximizes the scoring function. To apply this framework to a 2-layer setting, they propose a variational inference objective introducing an additional hidden proposal. The effectiveness of this method, tested on multiple datasets, is demonstrated in both 1-network and 2-network settings.

**Strengths:**

The proposed prompt optimization framework is both general and plausible, with its extension to the 2-layer setting using variational inference technically sound. The exploration of stacking multiple LLMs together is an intriguing and under-researched area. The paper's comprehensive experimentation on various datasets effectively showcases the method's applicability in both 1-layer and 2-layer settings.

**Weaknesses:**

The paper's title, "Deep Language Networks", is misleading, as the main content only employs 2-layer LLMs. The term 'deep' should be avoided to prevent any misconceptions and exaggeration.

While the introduction claims that existing modular approaches are heavily dependent on prompt engineering to break a task into smaller tasks, the proposed method also relies on an initial given prompt. This initial prompt prescribes the subtask for each layer. For instance, in the 2-layer setting experiments, the first layer always serves as the COT step, and the second layer provides the final answer. The roles of the two layers are determined by the initial human-provided prompt.

Though the paper suggests that it's easy to generalize from a 2-layer network to multiple layers, it's not straightforward to establish the initial prompts for each layer. While it's manageable in a 2-layer setting, mirroring the COT steps where the first layer handles reasoning and the second layer provides the final answer, it becomes substantially more challenging as the layer size increases. Therefore, the main motivation of this paper may not be as practically applicable as suggested.

**Questions:**

In the context of variational inference, is the LLM that parameterizes q(h) separate?

A straightforward baseline would be COT+APE. How does the performance of this simple baseline compare?

Stacking multiple LLMs increases computational complexity. An alternative is to directly use an ensemble of multiple LLMs. How does the proposed method compare with a simple ensemble of multiple LLMs? This could offer insights into whether LLMs should be combined horizontally or vertically.

**Limitations:**

Please refer to the Weaknesses section for a detailed discussion of the paper's limitations.

---

> ### Author Rebuttal · Authors · 2023-08-09
>
> Thank you for your review!
>
> ***“The paper's title, "Deep Language Networks", is misleading given that it’s only 2 layers”***
>
> Our title expressed the conceptual framework that originated the idea of the variational inference prompt optimization algorithm, which is our main contribution. While we acknowledge that our models are not “deep”, the algorithm can be readily extended to multiple layers, at least theoretically, as you can find in the Appendix. If the reviewer thinks it’s useful, we could either remove the mention of “Deep Language Networks” in the title and write “Two-Layer Language Networks”, or specify “Towards Deep Language Networks..”.
>
> ---
>
> ***“While the paper argues that modular methods are heavily dependent on prompt engineering to break a task into smaller tasks, the proposed method also relies on an initial given prompt.”***
>
> We agree! We rewrote our introduction and pointed out that our method is also relying on initial prompts and thus can be seen as a step towards **“integrating learnable components in a human-designed pipeline of prompts”**: in fact, we see our algorithm as a principled way to fine-tune initial prompts that were originally human generated.
>
> ---
>
> ***“The paper claims that it's easy to generalize from a 2-layer network to multiple layers”***
>
> We apologize for the confusion: we did not mean to claim that it is practically easy to optimize more than 2 layers: indeed, we are open in discussing the optimization difficulties encountered in the two layer case. The algorithm is readily extendable to multiple layers as we point out in the Appendix. We hope that our paper can be considered as a starting point from which we can think about training more than two layers in a principled way. We have rewritten our intro to make sure the training challenges with longer chains are clear.
>
> ---
>
> ***“It's not straightforward to establish the initial prompts for each layer.”***
>
> We added a discussion about this point in the paper. Engineering prompts is arguably easier than choosing initializations of random neural network weights: this has long remained a difficult question. When working with commercial applications and engineering libraries such as, e.g. LangChain, designers manually craft pipelines of prompts. One potential straightforward application of our prompt optimization algorithm is to fine-tune an already carefully human engineered chain of prompts given some task data.
>
> ---
>
> ***“COT+APE / Ensemble of multiple LLMs”***
>
> Thank you for suggesting the baselines. For CoT+APE, we followed the procedure described in Section 4.3 of [Zhou et al., 2023], that is we use APE to find a prompt starting with “Let’s” that maximizes the likelihood of correct chain-of-thought reasoning steps. We ran that baseline on the four BBH datasets and observed lower performance except for the date dataset. Looking at the generated prompts from CoT+APE, they are much simpler than what we can obtain with DLN. For instance, on hyperbaton: “Let’s review the correct adjective order in English” vs the example shown in appendix D. Additionally, we also report the scores obtained by a system ensembling three DLN-1, we train the three DLN-1 individually and ensemble by majority voting during test. These results can be found in ***Table A/B*** in the shared response.
>
> ---
>
> ***“In the context of variational inference, is the LLM that parameterizes q(h) separate?”***
>
> We use the same LLM (davinci-003) but in principle it can be separate, q(h) could either be a bigger LLM or a smaller LLM. Understanding the implication of this is definitely an interesting direction for future work.

---

> > ### Comment · Area_Chair_Lbk3 · 2023-08-16
> > **Any further questions ?**
> >
> > Dear SaT7,
> > Do you have any further clarifications you need from the authors ? Thanks.

---

> > > ### Comment · Reviewer_SaT7 · 2023-08-19
> > > **Thanks for the Rebuttal**
> > >
> > > I appreciate the detailed rebuttal you've provided. Considering the points raised, I would like to suggest that the paper title include the phrase "Towards Deep Language Networks..." to better encapsulate its focus. However, it's important to acknowledge that several of my concerns are contingent upon a substantial rewriting of the paper. Given the current state, I am inclined to maintain my current score and continue to view the paper as a borderline case.

---

### Official Review · Reviewer_Ttvx · 2023-07-07

**Soundness:** 3 good
**Presentation:** 3 good
**Contribution:** 3 good
**Rating:** 6
**Confidence:** 3

**Summary:**

The authors provide an interpretation of LLMs as shallow language networks. They explained how One-layer language networks can be used for joint prompt training, and then moved to stacked (two) language networks. The authors propose to use variational inference for the training, and figured out a few practical instantiation that makes the optimization work well.



**Strengths:**

Viewing LLMs as stochastic layers and prompts as tunable parameters, the authors provide a framework for joint prompt training of stacked LLMs using variational inference. The overall idea looks sound, and the framework and variational inference solution look elegant. The authors also experimentally verify their Deep Language Network on a few artificial datasets, and achieve comparable performance. The ablation study aligns with the authors' claim.

**Weaknesses:**

-- As also claimed by the authors, it is difficult to make Eq1 (the bound) tight, and the optimization is also not straight-forward. In order to make the whole system work, a lot of practical instantiation is needed, making the whole system more cumbersome. In order to get diverse prompts and identify the near-optimal prompts, I expect the value of K and N in algorithm 2 will not be too small. An ablation study looking into that, to study the computational cost and performance effect given different K and N could be interesting.

-- Also, this work only evaluates on small scale datasets, and works on some artificial tasks. This may prevent the audience from understanding the benefit of the proposed methods. Is it because the training/evaluation is not so efficient, so the authors decided to work on a small scale?

-- I wonder if the proposed methods can be easily extended to three layers or beyond. Technically, it seems possible, but the optimization could be very challenging.

-- In line 186, the authors mentioned that they selected lambda for each task. This could be cumbersome when working on a new task/dataset.

**Questions:**

Can the authors explain the results in Table 2 with richer information. Why does the approach seem to work very well on Nav and Subj, but worked clearly worse than other strong baselines? The 2-layer DLN end-to-end worked badly presumably due to the difficulty of optimization mentioned in section 4?

**Limitations:**

The authors present some bullet-points on potential limitations in the draft, I did not identify significant negative societal impacts.

---

> ### Author Rebuttal · Authors · 2023-08-09
>
> Thank you for your review and insights.
>
> ---
>
> ***“An ablation study to study the computational cost and performance effect given different K and N could be interesting.”***
>
> An ablation study would be absolutely interesting and we will add this to the camera ready. Empirically, we find that setting the number of prompt samples `N` 10/20 is enough. We found this number by optimizing dev. Hyperbaton performance. We suspect that more prompts won’t help due to diversity issues in the prompt proposal. Augmenting the hidden samples `K` is costly especially if every hidden layer generation gets long. We found `K` ~5/10 to work well across tasks. We found this number by optimizing Navigate performance.
>
> ---
>
> ***“Can the authors explain the results in Table 2 with richer information. Why does the approach seem to work very well on Nav and Subj, but worked clearly worse than other strong baselines? The 2-layer DLN end-to-end worked badly presumably due to the difficulty of optimization mentioned in section 4?”***
>
> Thank you for noticing this. For Date, we noticed a dataset problem, where the dev set is very small. Sometimes DLN-2 would get 90% on dev and 40% on test. For Logic.7, we think that the base model (dv003) is simply not strong enough to perform such reasoning and therefore we would need a "deeper" DLN where each step does a simpler reasoning step. Note that Logic.7 is the hardest of the logic datasets in BBH.
>
> ---
>
> ***“I wonder if the proposed methods can be easily extended to three layers or beyond. Technically, it seems possible, but the optimization could be very challenging.”***
>
> We agree. It can be very challenging. Notably, one the most challenging things is ensuring enough diversity in the prompt proposals. However, we think that in industrial applications, the optimization won’t start de novo, but from a set of carefully human designed prompts. In that setting, DLN can be considered as a way to fine-tune this well-thought initialization, in a principled way. Future work should be concerned in finding such scenarios which are pervasive in industry but less so in academia.

---

> > ### Comment · Reviewer_Ttvx · 2023-08-18
> >
> > Thanks for the information, and I agree that the planned changes would make the paper stronger, and I’m looking forwarding a later version. I would keep the score as it is for now.

---

### Official Review · Reviewer_2Jdr · 2023-07-28

**Soundness:** 1 poor
**Presentation:** 2 fair
**Contribution:** 1 poor
**Rating:** 4
**Confidence:** 3

**Summary:**

This paper introduces a novel discrete prompt tuning method. The paper is discussed under a scenario that regarding discrete prompt as the only tunable parameters while freezing all other model parameters. In this sense, the author focuses on getting optimized prompt from one LLM and feeding which as the input prompt to next LLM. Such kind of stacking LLMs workflow is regarded as the main contribution as the authors claimed. Moreover, the authors introduce a new latent variable to further improve this pipeline.

In the experiment, the authors verify the effectiveness of the proposed method via comparing with a bunch of baselines including several in-context learning method and one instruction-tuning method APE. The authors highlight the outperformance of their method over 0-shot GPT-4 is achieved via using text-davinci-003 backbone on Hyper, Tree, Disaster dataset.

**Strengths:**

originality: This paper shows a novel discrete engineering method from a new perspective that interprets the process of getting optimal prompts as the stack of two LLMs. This angle is novel and interesting.
From the methodology perspective, authors develop their method based on improved APE and generalize which to 2-layer case.

quality: overall quality is mediocre, though the idea is novel, the authors do not provide enough evidence to support their claim.

clarity: the authors attach the codes for 1-layer & 2-layer case, which makes it easier to interpret their ideas. However, the intuition for some settings need to be improved.

significance: authors evaluate the proposed 1-layer method on 9 tasks and 2-layer method on 4. They re-run the experiment 3 times for verify the significance.



**Weaknesses:**

1. The paper is not comparing with enough correct baselines. The authors are mainly comparing with 0-shot GPT3&4 and APE method in two main tables. However, if you regard the number of training example that proposed method can touch with as the x-shot, then the comparison objective should be x-shot GPT3&4 instead of 0-shot GPT3&4 (especially in Table 2). It's widely acknowledged that increasing the number of few-shot examples (in a range) would enhance model's performance. Is it still fair to compare with 0-shot model with your proposed method? Then it means, the only baseline authors are correctly comparing with is APE. I'd encourage authors to compare more baselines in both discrete prompt engineering and continual-space prompt tuning area.

2. The generalization of the proposed method is questionable. The author claims their proposed method can generalize to more number of layers. But the comparison between 2-layer and 1-layer model is insufficient. Authors only compare them on 4 datasets and on some of them the 1-layer results are even better than 2-layer's. It's not convincing.

3. Even the authors have mentioned in conclusion&future work that testing with other LM is needed. I want to highlight this as a major weakness of the paper. The value of this paper should be reevaluated is it only works on GPT-3 model. Why not stacking two GPT-4 models which should not be hard to do (just change slightly in API calling)? Or maybe on BERT/RoBERTa models. Without corresponding evidence, it's hard to justify the effectiveness of the proposed method.

4. The intuition and some setting of the proposed method is unclear. For example, in Line 137, the authors suddenly define the a weight without illustrating why the sum of these two terms is a good fit for their purpose. I am little confused to interpret.

5. The authors also mentioned that they hope to match the performance of the largest LLMs without incurring the large computational and data cost. However, one should also be careful that GPT-3 API calling is also a considerable computational cost. Without disclosing the number of times they call the GPT-3 API, it's hard to support their claim as incurring low computational cost.

**Questions:**

1. I kind of curious why you are adopting a subset of test set for evaluation. Any reason for that?
2. I kind of doubt the performance of 0-shot GPT-4. For example, on Subj dataset, FT RoBERTa and prompt tuning RoBERTa can achieve ~97% score, while 0-shot GPT-4 can only get 65.8. I am little shocked by GPT-4's bad performance on this task. It would be better for the authors to release the prompt of 0-shot gpt-4 they use for reproducing purpose.
3. Why not reporting the 95% confidence interval on APR & 1-Layer LN & 2-Layer LN in Table 2?
4. Why you split the main results into two tables?

**Limitations:**

typo:
Algorithm 2 Line 13 \beta_k -> \beta_1
Algorithm 2 Line 18 \pi_0^n -> \pi_0^I
Algorithm 2 Line 19 \pi_1^n -> \pi_1^I

---

> ### Author Rebuttal · Authors · 2023-08-09
>
> Thank you for your review!
>
> Please find hereafter our answers:
>
> ---
> ***“1. The paper is not comparing with correct enough baselines. [...] it means, the only baseline authors are correctly comparing with is APE.”***
>
> In the paper, in addition to APE, we compared to other non 0-shot baselines: ICL (5-shot) uses 5 examples from the training set, so we have one additional baseline which is *not* zero-shot. KATE retrieves from 400 examples, therefore the same number as we use for optimization, and uses 5 of those to perform few-shot learning.
>
> For completeness, we performed additional experiments with 10-shots and 32-shots on the tasks we compared with DLN-2 and the results are as follows. Please, take a look at the shared response (***Table A***) for these results.
>
> As reported in the above table, for some tasks (e.g., Logic 7), the length of extra in-context demonstrations could quickly use up the context length limit (4096 in our case of text-davinci-003). This suggests an advantage of prompt optimization in comparison to in-context learning, despite both methods leverage information from the training set.
>
> ---
>
> ***“2. Scale beyond two layers; two layers is not always better than one layer”***
>
> Scaling beyond two layers might be a hard problem. We think that prompt optimization can benefit from human engineered initializations, which can make optimization easier. We hope that our algorithm can serve as a starting point to fine-tune human crafted chains of prompts.
> Some tasks might not benefit from CoT or reasoning paths. For this paper, we ultimately consider the 1 vs 2 layers as a model selection problem for each task.
>
> ---
>
> ***“3. Testing with other LM is needed. I want to highlight this as a major weakness of the paper.”***
>
> We tested our prompt optimization algorithm both with GPT-4 and an open source model WizardLM. We cannot unfortunately stack two GPT-4 models, as we need to have access to the log probabilities, which are not available at this time. We couldn’t run 2 layers on the open source model given that we require “echo” functionality of the log_probs and VLLM (the service we use for inference) doesn’t have it yet. This requires more time. We will report the numbers in the camera ready.
>
> We report mean accuracy across 3 seeds and cost for inference over the test set as the number of processed tokens (less is better). These results were obtained using the latest GPT-4 openai endpoint. These results can be found in ***Table C*** and ***Table D*** the shared response.
>
> ---
>
> ***“5. the authors suddenly define the a weight without illustrating why the sum of these two terms is a good fit for their purpose.”***
>
> The sum of these two terms corresponds to weighting each sampled hidden state by their probability under the posterior distribution. This can be viewed as a biased but more robust importance weighting scheme where the prior distribution is assumed to be uniform. We will make this clearer in revision.
>
> ---
>
> ***“6. Without disclosing the number of times they call the GPT-3 API, it's hard to support their claim as incurring low computational cost.”***
>
> For reference, the costs for calling GPT-3 are in the appendix. While DLN-2 requires twice as much compute as a single language layer using the same base LLM, our hope is that each language layer in a DLN-2 could be much smaller than the language layer in DLN-1 for the same performance, thus limiting or even negating the additional inference cost of using multiple layers.
>
> ---
>
> ***“Why you are adopting a subset of test set”***
>
> Adopting a subset of the test set is also a common practice shared by other works due to computational costs associated with running large LMs. For example, this paper also uses a subset of the test set for evaluation: https://arxiv.org/abs/2104.08786
>
> ---
>
> ***“Doubt the performance of 0-shot GPT-4... FT RoBERTa and prompt tuning RoBERTa can achieve ~97% score”***
>
> For the 0-shot prompt, we use the instruction in the Appendix. FT and prompt tuning use gradient-based optimization and thus achieve a higher score. The following paper shows the superiority of gradient-based optimization wrt 0-shot and in-context learning for NLP tasks https://arxiv.org/abs/2205.05638. Our new results use the latest GPT-4 version available on the API, the score improved to 74.4% with this new model but still significantly far from ~97% score that you suggest.
>
> ---
>
> ***“Typo in Algorithm 2”***
>
> We fixed the typo in Algorithm 2, line 17-18 $n$ should be $N$, thanks for noticing!

---

> > ### Comment · Reviewer_2Jdr · 2023-08-13
> > **further improvements are necessary for this paper**
> >
> > Thank you for your reply. Here are a few thoughts I'd like to share regarding your responses.
> >
> > 1. **"Insufficient comparison with correct baselines"**
> >
> > Table A indeed reveals the substantial benefit of incorporating more examples, resulting in a notable improvement of approximately +10 for both the ICL and KATE models in navigation and subjectivity tasks. As previously pointed out, it's worth contemplating whether this comparison is equitable, given that you extract information from 400 examples while the baselines have access to only 5/10/32 examples. To promote thorough analysis, I recommend conducting an ablation study where your models are trained with the same number of examples as the baselines.
> >
> > Additionally, I reiterate my belief that the current version lacks an adequate comparison with a comprehensive range of baselines. Expanding the scope of baselines for comparison could enhance the robustness of your findings.
> >
> > 2. **"Scale beyond two layers; two layers is not always better than one layer"**
> >
> > You've highlighted a significant discrepancy in your response. While you explained that "scaling beyond two layers might be a challenging issue," your submission predominantly emphasizes the ability of your model to incorporate numerous LLMs and attains superior performance compared to a single layer (as evidenced between Line 3-5). This might be perceived as an overstatement. Notably, your original submission omits any mention of the unfavorable generalization to additional layers.
> >
> > 3. **"Testing with other LM is needed. I want to highlight this as a major weakness of the paper."**
> >
> > You've made a valid point regarding the inclusion of GPT-4 and the open-source model WizardLM, which is a positive addition. However, I find it difficult to concur with the methodology of computing solely based on the number of tokens during inference. Indeed, your model necessitates training, which involves multiple API calls, unlike ICL which doesn't require training. To ensure a fair comparison, wouldn't it be more appropriate to report the number of tokens involved in both stages? Furthermore, I'd like to emphasize the importance of presenting comprehensive results across a broader range of datasets, rather than solely focusing on the four datasets mentioned earlier.
> >
> > **"Why you are adopting a subset of test set"**
> >
> > Utilizing a subset of the test set for evaluation becomes more reasonable when evaluating the entire test set is resource-intensive or when baseline methods themselves utilize such an approach. However, in your case, neither of these circumstances seem to apply. Without aiming to be overly critical, I'd like to point out that the paper you referred to has two distinctions from your submission: firstly, it lacks a direct relationship to your work, and secondly, it employs a subset of the development set rather than the test set.
> >
> > I still not get the response for my questions **"Why not reporting the 95% confidence interval on APR & 1-Layer LN & 2-Layer LN in Table 2?"** and **"Why you split the main results into two tables?"**.
> >
> > Considering the current quality of the submission and the need to incorporate more comprehensive experiments, I maintain my stance that further improvements are necessary for this paper.

---

> > > ### Author Response · Authors · 2023-08-14
> > > **clarifications on your responses**
> > >
> > > Thank you for the answer. Before replying, we want to emphasize that these models are expensive and we need to be mindful when selecting which experiments to run. We would love it if you could provide a bit more information about what experiments you would like us to prioritize, which conclusion you would get from the additional results, and how that would affect your assessment of our work. While waiting for an answer, we still decided to run as many of the requested experiments as we could, hoping that this can address your concerns.
> > >
> > > ---
> > >
> > > ***As previously pointed out, it's worth contemplating whether this comparison is equitable, given that you extract information from 400 examples while the baselines have access to only 5/10/32 examples.***
> > >
> > > KATE consists of a retrieval step and a prediction step. The retrieval step is executed using *400* examples. Due to the limit of the context length in davinci-003, the prediction step can take at most 32 examples. Therefore, the KATE baseline we report uses all of 400 examples.
> > >
> > > We would be happy to implement other baselines you think are more equitable.
> > >
> > > ---
> > >
> > > ***the current version lacks an adequate comparison with a comprehensive range of baselines. Expanding the scope of baselines for comparison could enhance the robustness of your findings.***
> > >
> > > We would be happy to implement other baselines you suggest that can enhance the robustness of our work.
> > >
> > > ---
> > >
> > > ***To promote thorough analysis, I recommend conducting an ablation study where your models are trained with the same number of examples as the baselines.***
> > >
> > > We currently ran DLN-1 on the 4 tasks from the rebuttal and updated our results:
> > >
> > > | | nav / #tok | date / #tok | logic 7 / #tok | subj / #tok |
> > > | - | - | - | - | - |
> > > | ICL - 10-shot	| 61.3 / 151k | 62.9 / 263k | 38.9 / 529k | 72.0 / 119k |
> > > | ICL - 32-shot | 66.0 / 449k | 63.5 / 786k | exceeds ctx len | 83.2 / 343k |
> > > | DLN-1 - 32-shot | 69.1 / 26k | 55.0 / 136k | 42.1 / 77k | 83.1 / 55k |
> > >
> > > ---
> > >
> > > ***While you explained that "scaling beyond two layers might be a challenging issue," ... your original submission omits any mention of the unfavorable generalization to additional layers.***
> > >
> > > We do not think that “scaling beyond two layers might be challenging” means “unfavorable generalization to additional layers”. Instead, while it might be hard to train multiple layers, very much like it was hard for neural networks at the beginning, the potential gains could also be very real if one manages to do so. Our experiments with DLN-2 show that this could be the case. As mentioned in the general response, we will tone down the abstract and intro in the camera-ready.
> > >
> > > ---
> > >
> > > ***However, I find it difficult to concur with the methodology of computing solely based on the number of tokens during inference.***
> > >
> > > Please refer to the Appendix where we report the cost of training DLN-2. It costs 60 dollars to train a dln-1 and 273 dollars to train a dln-2. We therefore believe such amounts during training are thus negligible when serving the model for a large number of users: tokens spent per user request at test time is more important in this context, as this is a cost you can't amortize.
> > >
> > > ---
> > >
> > > ***Furthermore, I'd like to emphasize the importance of presenting comprehensive results across a broader range of datasets, rather than solely focusing on the four datasets mentioned earlier.***
> > >
> > > We restricted ourselves to the datasets reported because every run is fairly expensive. Please suggest datasets you find particularly useful to strengthen our work.
> > >
> > > ---
> > >
> > > ***“it employs a subset of the development set rather than the test set.***
> > >
> > > The reported results in Lu et al. are computed on a 256 subsample of the dev set. Please refer to Table 2 caption – first line – and their github repo: https://github.com/yaolu/Ordered-Prompt/blob/e93f70d0a5f6a8cfcadf6f2917c26eed265cd0be/config/cb.yaml#L2 The reason the authors give is for computational efficiency.
> > >
> > > ---
> > >
> > > ***"Why you split the main results into two tables?".***
> > >
> > > Because running DLN-2 is more expensive and we thought it would be sufficient to do it on 4 datasets, and we feel these 4 datasets are the most representative of the range of behaviours we want to analyze.
> > >
> > > ---
> > >
> > > ***"Why not reporting the 95% confidence interval on APR & 1-Layer LN & 2-Layer LN in Table 2?"***
> > >
> > > |                         | **Nav**        | **Date          | **Logic 7**    | **Subj**       |
> > > |:-----------------------:|:--------------:|:---------------:|:--------------:|:--------------:|
> > > | **0-shot**              | 64.1           | 56.4            | 45.9           | 61.7           |
> > > | **APE-15**              | 67.3 $\pm$ 7.7 | 32.1 $\pm$ 28.6 | 45.5 $\pm$ 4.7 | 61.3 $\pm$ 7.2 |
> > > | **APE-400**             | 56.9 $\pm$ 32.9| 23.5 $\pm$ 14.1 | 45.6 $\pm$ 12.4| 63.7 $\pm$ 9.2 |
> > > | **DLN-1**               | 67.1 $\pm$ 7.6 | 55.7 $\pm$ 4.5  | 47.5 $\pm$ 2.1 | 83.2 $\pm$ 6.5 |
> > > | **DLN-2 (end 2 end)**   | 83.1 $\pm$ 24.7| 65.9 $\pm$ 4.0  | 45.7 $\pm$ 3.5 | 85.9 $\pm$ 8.7 |

---

> > > > ### Comment · Reviewer_2Jdr · 2023-08-16
> > > >
> > > > I know that conducting experiments is expensive for sure. However, this shouldn't be a reason for preventing you do some meaningful experiments if they are necessary for your paper.
> > > >
> > > > **32-shot ablation study**
> > > >
> > > > That's precisely in line with my understanding. When considering an equal number of accessible data points, the proposed approach manages to outperform ICL baselines on just one dataset (also other baselines are missing here, the performance of other methods' 32-shot versions compared to your model remains uncertain).  If this holds true, can I deduce that your method excels over ICL only for one dataset while employing fewer tokens (thus, incurring fewer costs)?
> > > >
> > > > However, in a broader sense, can we confidently recommend your method to users seeking optimal performance whenever they encounter a new dataset? I'm concerned that the answer might be negative.
> > > >
> > > > **The claim about generalizing to more than 2-layer networks**
> > > >
> > > > Indeed, it's hard to overlook the fact that your initial submission's assertion could be seen as somewhat misleading, a sentiment shared by Reviewer SaT7. Moving forward, a more cautious approach to highlighting the capabilities of the proposed method might be advisable, particularly since it hasn't been subjected to testing involving more than two layers. This could contribute to a more accurate representation of the method's potential in your upcoming version.
> > > >
> > > > **Costs in training/testing**
> > > > Despite your assertion that training costs are a relatively minor factor, it's noteworthy that the training cost for the proposed 2-layer and 1-layer methods is 2.9 million and 1.4 million respectively, on a single dataset (in contrast, ICL-5-shot GPT-4 costs 83k, 291k, 66k on three test sets). When these training costs are combined with practical testing expenses, the overall expenditure does indeed largely exceed that of the baseline methods.
> > > >
> > > > I do see your point that testing takes majority part when the number of the test is larger. However, in your current situation, wherein the number of test set is small and the total cost of your model surpasses the baselines. Moreover, the separation of reporting training costs and emphasizing testing costs might be perceived as inequitable, especially for methods that do not entail training costs, even if the focus is primarily on testing expenditures.
> > > >
> > > > **Why you are adopting a subset of test set**
> > > > I read the paper and I know what they did. But you were still not explaining why you test under a subset instead of full set. As training on 2.9 million tokens takes roughly $273, according to table C the ICL-5-shot GPT-4 takes 66k and DLN-1 takes 57k on subj dataset. Given this context, it raises the question of whether the expense of testing on the entire subj test set would truly be prohibitively expensive.
> > > >
> > > >  **Why splitting two tables**
> > > > I just feel the two tables are very duplicated as they are comparing under the same datasets for the same purpose.
> > > >
> > > > ----
> > > > **I am summarizing my review comments for this submission as a conclusive evaluation. There is no need for a response.**
> > > >
> > > > I find it challenging to discern the advantages of employing the proposed method, especially when considering the scenario of controlling the same number of training data accessible. Firstly, traditional trainable methods such as fine-tuned Roberta showcased notably superior performance in some of the tasks tackled by the authors. Secondly, in instances where GPT API calls are exploited, the outcomes do not exhibit a distinct pattern compared to baselines that also employ blackbox models.
> > > >
> > > > Certain aspects of the submission appear misleading (as discussed in my previous response) — notably the claim about stacking l-layer LM, and the presentation choice of not combining training and testing costs.
> > > >
> > > > The experimentation aspects require greater rigor. While the authors included numerous additional experiments during the rebuttal, some were conducted on partial datasets rather than the complete ones specified in the paper. I hope the authors will incorporate and conduct experiments on all nine datasets mentioned in their next version, rather than solely for the purpose of persuasion. Furthermore, some of the baselines' implementations could benefit from enhanced rigor. For instance, the authors mentioned that the ICL method randomly samples five examples from the training set in Line 226. Given our understanding that the order and quality of ICL examples influence results, conducting a significant test considering the randomness involved would be prudent.
> > > >
> > > > **In conclusion, I believe this submission is not yet prepared for acceptance.**

---

### Author Rebuttal · Authors · 2023-08-09

We thank the reviewers for their comments. In addition to the replies to the specific questions raised by each reviewer, we want to address points that were raised multiple times.

---

***About the discrepancy between the ambition of our proposed framework and its technical instantiation***

While the framework can technically accommodate more than two layers, we acknowledge that we limit its instantiation to 1 and 2-layer networks. Some of the barriers to extending this framework to more general architectures, like the quality of the variational inference, will likely be challenging. We will rephrase our claims and be more explicit about the difference between the theoretical approach and what has currently been tested. While we believe this framework is both conceptually interesting and has already delivered some interesting results, we do not want to downplay the challenges to overcome to instantiate it more generally, nor the limitations of our current analysis, mainly due to the large computational costs of working with models of this scale.

---

***Regarding the lack of comparison with more models and baselines***

In addition to reemphasizing that we did compare with several prompt engineering and few-shot techniques in the original submission, we added comparisons to more models, also using different LLMs as language layers, as allowed by our framework. Particularly we added results for:
- APE+CoT
- ICL (in-context learning) and KATE with more in-context examples
- DLN-1 ensemble
- DLN-1 on top of GPT-4
- DLN-1 for open-source models such as [WizardLM-13b-v1.2](https://github.com/nlpxucan/WizardLM), an open sourced LLM developed based on Llama-2.

We cannot run DLN-2 on top of GPT-4 due to the lack of available log-probabilities. We couldn’t compute DLN-2 for open-source models in time for the rebuttal, but we will report the results in the camera ready version. We also report the cost (lower is better) in terms of total number of tokens for the test set (prompts included, #tok). We emphasize the cost at the testing time because it is more relevant in real-world deployment and the training cost is one-off. Note that: 1/ We used the newest GPT-4 version (0613) in Table C; 2/ we re-optimized DLN-2 E2E hyper-params on the datasets tested and got improvements, especially for navigate.

Here are all the results:

---

| ***Table A*** | nav | date | logic 7 | subj |
| - | - | - | - | - |
| ICL - 5-shot | 56.5 | 62.1 | 36.7 | 76.4 |
| ICL - 10-shot | 61.3 | 62.9 | 38.9 | 72.0 |
| ICL - 32-shot | 66.0 | 63.5 | exceeds ctx len | 83.2 |
| KATE - 5-shot | 56.9 | 61.1 | 44.4 | 71.1 |
| KATE - 10-shot | 59.5 | 62.0 | 41.6 | 73.9 |
| KATE - 32-shot | 67.5 | 62.8 | exceeds ctx len | 80.4 |
| DLN-1 | 67.1 | 55.7 | 47.5 | 83.2 |
| DLN-2 | 82.9 | 63.9 | 45.2 | 85.9 |
| DLN-1 Ensemble-3-seeds | 65.6 | 51.2 | 46.8 | 83.2 |

***TLDR***: For Text-Davinci-003, increasing the number of ICL examples helps performance, but cannot match DLN-1, DLN-2 performance. More in-context examples could quickly use up the context length limit.

---

|  ***Table B*** | hyper | nav | date | logic 7 |
| - | - | - | - | - |
| APE-15 |  68.5 ± 5.5 | 67.3 ± 7.7 | 32.1 ± 28.6 | 45.5 ± 4.7 |
| CoT+APE | 50.9 ± 0.8 | 61.5 ± 1.5 | 58.6 ± 2.2 | 38.9 ± 1.6 |
| DLN-1 | 91.9 ± 3.0 | 67.1 ± 7.6 | 55.7 ± 4.5 | 47.5 ± 2.1 |
| DLN-2 | - | 82.9 | 63.9 | 45.2 |

***TLDR***: DLN-1 and DLN-2 outperform CoT+APE.

---

| ***Table C*** | nav / #eval tok | logic 7 / #tok | subj / #tok |
| - | - | - | - |
| 0-shot GPT-4 | 73.8 / 16k | 62.5 / 50k | 74.4 / 13k |
| ICL-5-shot GPT-4| 72.6 / 83k | 60.0 / 291k | 90.4 / 66k |
| ICL-16-shot GPT-4| 77.1 / 233k | 62.2 / 822k | 93.8 / 181k |
| DLN-1 GPT-4| 78.5 / 86k | 64.0 / 80k | 90.6 / 57k |

***TLDR***: DLN-1 improves over ICL on 2 out of 3 tasks on GPT-4 at a comparable token cost. Some tasks do not benefit from ICL (i.e. reasoning tasks) while other tasks like SUBJ benefit significantly. In reasoning tasks, DLN-1 outperforms ICL even for GPT-4 by learning strategies to solve the problem. We combined DLN-1 and 5-shot ICL for SUBJ and matched 16-shot performance at a lower token cost, i.e. DLN-1 GPT-4 + 5-shot: 93.8 / 116k

---

| ***Table D*** | nav / #eval tok | logic 7 / #tok | subj / #tok |
| - | - | - | - |
| 0-shot WizardLM-v1.2 13B | 58.0 / 32k | 0.0 / 54k | 65.8 / 14k |
| ICL-5-shot WizardLM-v1.2 13B| 56.0 / 240k | 28.0 / 318k | 50.8 / 74k |
| DLN-1 WizardLM-v1.2 13B | 61.1 / 48k | 31.0 / 69k | 79.8 / 17k |

***TLDR***: Open models seem significantly less able to capture few-shot examples in the context. DLN-1 outperforms ICL on all tasks here.

---

### Author Response · Authors · 2023-08-16
**Questions/Concerns?**

Dear Reviewers,

Thank you for your time. We know it’s a busy moment, but we would like to know if there’s anything else we can do to clarify your concerns about our paper.

---

***Reviewers Y4LZ, Ttvx:***
Thank you for your positive feedback, we hope the rebuttal helped in answering your questions.

---

***Reviewer SaT7:***
- Your first concern was the paper contained “deep” in the title while only dealing with 2-layers DLN. We are open to changing the title, and we already toned down the abstract and intro by making the limits of this work to only 2-layers more evident. Would you have suggestions for the title?
- We also implemented your proposed APE+CoT baseline as you can find in the common response. Was that helpful in addressing your concerns?

---

***Reviewer 2Jdr:***
Thank you for engaging with us. If you could be more specific about which baselines and which other datasets you think could robustly improve our work, we would be happy to work on that, always in the limit of our computational resources. Please note that we also added a APE+CoT baseline following SaT7’s suggestion.

---

Is there anything else that we can do to address your concerns?

Thanks!

---

### Author Response · Authors · 2023-08-17
**Common response: abstract update and clarifying test set sizes**

Dear reviewers, we wish to provide both an update and a clarification.

## Abstract update

We updated the abstract to align it better with the current state of the framework. We are confident it will address the issues several of you raised. The new abstract is:

We view large language models (LLMs) as stochastic language layers in a network, where the learnable parameters are the natural language prompts at each layer. We stack two such layers, feeding the output of one layer to the next. We call the stacked architecture a Deep Language Network (DLN). We first show how to effectively perform prompt optimization for a 1-Layer language network (DLN-1). We then show how to train 2-layer DLNs (DLN-2), where two prompts must be learnt. We consider the output of the first layer as a latent variable to marginalize, and devise a variational inference algorithm for joint prompt training. A DLN-2 reaches higher performance than a single layer, sometimes comparable to few-shot GPT-4 even when each LLM in the network is smaller and less powerful.


## Dev / Test set confusion

After discussion, we felt maybe some points regarding test set size were still unclear and thus we would like to clarify.

***Why don’t we use the full test set?***

The test sets of the original data are below:

| Task | Test Examples |
|:------:|:------:|
| Hyperbaton | 250 |
| Navigate | 250 |
| Date Understanding | 250 |
| Logic. 7 | 250 |
| Subj | 2000 (8x) |
| Mpqa | 2000 (8x) |
| Trec | 500 (2x) |
| Disaster | 5430 (~22x) |
| Airline | 7319 (~29x) |

Given that we have a large set of baselines, ablation study settings and 3 seeds, we would like to be mindful of the number of queries to GPT API. We chose to subsample every test set to use 250 examples.

We would like to stress that we are not the first in doing so. This closely follows the setting in Lu et al. (ACL 2022 outstanding paper). They report:

> For evaluation, we sub-sample 256 samples of the validation sets for all datasets to control for the GPT-3 inference costs as it requires the usage of a monetary paid-for API.

This means they sample 256 data points from the dev set and ***report scores on those 256 data points***.. We can confirm this from [their official code](https://github.com/yaolu/Ordered-Prompt/blob/main/config/subj.yaml#L2). We believe this could have caused the confusion because Lu et al. do not conduct a hyperparameter search and directly use the 256 dev data points for testing.

---

### Decision · Program_Chairs · 2023-09-21

**Decision:**

Accept (poster)

**Comment:**

The paper proposes to do prompt engineering for a single language model and a stack of two language models. The former is just done through a Monte Carlo sampling (sample prompts which are modifications of other prompts using the language model itself, and then chose the one with the best scores) and search type strategy; the latter is done by a variational technique. In both cases the search is over the discrete prompts.

They show results that are sometimes better than the baselines; but they admit that a couple of the datasets have issues (For Date, we noticed a dataset problem, where the dev set is very small. Sometimes DLN-2 would get 90% on dev and 40% on test. For Logic.7, we think that the base model (dv003) is simply not strong enough to perform such reasoning and therefore we would need a "deeper").

The reviewers didn't like it that they called it Deep Language Networks because it wasn't really deep. Others didn't like it because they thought the paper didn't do enough ablation experiments, in their opinion. They also thought that one of the reasons this method did better was that it saw a lot more data in the samples. The authors ended up doing a lot of the experiments that the authors asked for. But it complicates the story and how good the paper turns out on how the presentation of all these new experiments sits with the rest of the paper. Nevertheless the method presented is novel, and gives some rigor to the prompt-engineering process. And the results, while mixed some times, were indicative that the method could indeed work. Authors did perform a lot of the ablations that the reviewers asked for, and I hope the authors will do a decent job of incorporating them.